# Murine blastocysts generated by in vitro fertilization show increased Warburg metabolism and altered lactate production

Seok Hee Lee[1], Xiaowei Liu[1], David Jimenez-Morales[2], Paolo F Rinaudo[1]*

[1]Center for Reproductive Sciences, Department of Obstetrics and Gynecology, University of California, San Francisco, San Francisco, United States; [2]Division of Cardiovascular Medicine, Department of Medicine, Stanford University, Stanford, United States

*For correspondence:
rinaudop@obgyn.ucsf.edu

Competing interest: The authors declare that no competing interests exist.

**Abstract** In vitro fertilization (IVF) has resulted in the birth of over 8 million children. Although most IVF-conceived children are healthy, several studies suggest an increased risk of altered growth rate, cardiovascular dysfunction, and glucose intolerance in this population compared to naturally conceived children. However, a clear understanding of how embryonic metabolism is affected by culture condition and how embryos reprogram their metabolism is unknown. Here, we studied oxidative stress and metabolic alteration in blastocysts conceived by natural mating or by IVF and cultured in physiologic (5%) or atmospheric (20%) oxygen. We found that IVF-generated blastocysts manifest increased reactive oxygen species, oxidative damage to DNA/lipid/proteins, and reduction in glutathione. Metabolic analysis revealed IVF-generated blastocysts display decreased mitochondria respiration and increased glycolytic activity suggestive of enhanced Warburg metabolism. These findings were corroborated by altered intracellular and extracellular pH and increased intracellular lactate levels in IVF-generated embryos. Comprehensive proteomic analysis and targeted immunofluorescence showed reduction of lactate dehydrogenase-B and monocarboxylate transporter 1, enzymes involved in lactate metabolism. Importantly, these enzymes remained downregulated in the tissues of adult IVF-conceived mice, suggesting that metabolic alterations in IVF-generated embryos may result in alteration in lactate metabolism. These findings suggest that alterations in lactate metabolism are a likely mechanism involved in genomic reprogramming and could be involved in the developmental origin of health and disease.

## Editor's evaluation

Some children conceived by assisted reproductive technologies (ART) exhibit metabolic differences compared to those conceived naturally and the causes are unknown. This work reveals possible explanations for the metabolic differences and provides opportunities to improve ART and prevent the differences. This is a valuable contribution and will be of special interest to practitioners of ART, as well as to developmental and reproductive biologists.

## Introduction

Assisted reproductive technologies (ART), including in vitro fertilization (IVF) show excellent success resulting in the birth of over 8 million children worldwide (*Berntsen et al., 2019*). While the great majority of children born with these technologies are healthy, it has also been suggested that ARTs

could be associated with an increased risk of adverse perinatal outcome and possibly an increased predisposition to adult-onset chronic diseases (*Berntsen et al., 2019*; *Kleijkers et al., 2014*; *Qin et al., 2017*). In particular, several reports have shown that ART offspring display altered growth pattern (*Ceelen et al., 2009*; *Kleijkers et al., 2016*), increased risk of cardiometabolic dysfunction (*Ceelen et al., 2008*; *Guo et al., 2017*), higher systolic and diastolic blood pressures (*Cui et al., 2021*; *Guo et al., 2017*; *Meister et al., 2018*), premature vascular aging (*Meister et al., 2018*), altered lipid metabolism (*Guo et al., 2017*), and increased the prevalence of cerebral palsy (*Goldsmith et al., 2018*). Therefore, assessing the health of ART offspring and understanding the safety of ART is of critical importance.

Importantly, different culture media and oxygen concentrations may result in different stress to the embryo (*Chronopoulou and Harper, 2015*; *Feuer et al., 2016*; *Rinaudo and Schultz, 2004*; *Rinaudo et al., 2006*) with possible subsequent alterations in health of the individual. In particular, either 5% (physiologic) or 20% oxygen (atmospheric) concentrations have been widely used in clinical settings, and in 2014 only 25% of 265 IVF clinics reported of using 5% oxygen for culture of human embryos (*Christianson et al., 2014*). This is relevant, since it is well known that environmental oxygen can influence embryo development and their intracellular redox balance (*Bavister, 2004*; *Rinaudo et al., 2006*). Overall, low oxygen tension can significantly improve the cleavage rate, implantation, pregnancy and birth rates in humans compared with an atmosphere condition (*Adam et al., 2004*; *Bavister, 2004*; *Ciray et al., 2009*; *Ma et al., 2017*; *Meintjes et al., 2009*; *Preis et al., 2007*; *Waldenström et al., 2009*). Supraphysiological oxygen concentrations result in an increase in reactive oxygen species (ROS) (*Ryan et al., 1998*) and likely result in oxidative damage and metabolic alterations to embryos.

Despite investigations (*Gardner and Wale, 2013*), it is unknown how metabolism is reprogrammed in early embryos following culture ex vivo. This is a critical question, since metabolic alterations occurring during embryonic development will likely be responsible for epigenetic reprogramming; therefore, understanding these metabolic alterations might provide an initial window into what ultimately will determine an individual's health (*Tzika et al., 2018*). Preimplantation embryo metabolism is unique and different from somatic cells. The Krebs cycle is the main source of energy throughout the preimplantation period. Glycolysis is slowly utilized during the first 1–2 days of development but an increase in glycolysis and oxygen consumption (via mitochondria) is notable at the blastocyst stage (*Leese, 2012*). Importantly, preimplantation embryos show an accentuation of the Warburg metabolism (*Warburg, 1956*), i.e., higher production of adenosine triphosphate (ATP) through glycolysis and increased conversion of pyruvate into lactate, even in the presence of oxygen (*Redel et al., 2011*). This process is facilitated by the lactate dehydrogenase class of enzymes (LDH-A and LDH-B) (*Redel et al., 2011*).

Given our result to date, we sought to obtain a clearer understanding of embryonic metabolism in mouse embryos generated in vivo or in vitro. Blastocysts generated in vivo after mating were used as a control (flushed blastocyst group or FB). Two experimental groups were used: embryos generated by IVF and cultured to the blastocyst stage in optimal conditions using potassium simplex optimization medium (KSOM) medium with amino acids and 5% oxygen concentration (IVF 5% or optimal conditions group) or 20% oxygen concentration (IVF 20% or suboptimal conditions group). These groups of embryos were compared for: (1) oxidative stress (including ROS levels, markers of oxidative damage to embryos, and glutathione level); (2) changes in mitochondria/glycolytic function using the Agilent Seahorse XF HS Mini Analyzer and measuring selected metabolites (lactate and pyruvate) and embryonic pH; and (3) comprehensive proteomic analysis.

We found that embryos generated by IVF show profound alterations in metabolism. These findings provide a mechanistic insight that links culture conditions to metabolic states and might be useful to select embryos with greater viability.

## Results

### Embryo generated by IVF show similar morphology but lower development to the blastocyst stage and increased in ROS and oxidative damage

Only expanded blastocysts of similar morphology were used for all experiments (*Figure 1—figure supplement 1A-C*). Overall, IVF generated embryos showed lower development if cultured with atmospheric oxygen (*Figure 1—figure supplement 1D,E*). IVF generated blastocysts had lower total cell number, compared to in vivo generated blastocysts (*Figure 1—figure supplement 1F*). These morphologic and developmental changes were mirrored by increase in oxidative stress in IVF-conceived embryos, with embryos exposed to $20\%O_2$ (IVF $20\%O_2$) showing higher oxidative stress.

In particular, IVF-generated blastocysts had significantly decreased level of GSH compared with in vivo blastocysts (*Figure 1A and B*), increase in intracellular ROS (*Figure 1C*) and extracellular ROS levels (*Figure 1D*).

Importantly, IVF-generated embryos showed increased in oxidative damage to DNA (intensity of 8-OHdG, *Figure 2A and B*), lipids (PGF2-alpha, *Figure 2C and D*), and proteins (levels of DNPH, *Figure 2E and F*).

### IVF generated blastocysts showed decreased mitochondrial activity and increased glycolytic function, indicative of increase Warburg metabolism

We (*Belli et al., 2019*) and others *Acton et al., 2004*; *Ren et al., 2015* have shown that IVF-generated blastocysts show an increase in mitochondrial alterations. To evaluate the physiological consequences of increase in oxidative stress to embryos, mitochondrial metabolism (oxygen consumption rate, OCR), and glycolysis (proton efflux rate, PER; extracellular acidification rate, ECAR) were assessed using the Agilent Seahorse XF HS Mini (*Figure 3*). IVF-derived blastocysts showed significantly lower basal steady-state OCR (*Figure 3A and B*), ATP production (*Figure 3C*) and maximal respiration (*Figure 3D*), compared to in vivo-conceived control. Overall, these effects were accentuated in embryos cultured in atmospheric oxygen (IVF$20\%O_2$ group).

Analysis of glycolytic function revealed that IVF-generated embryos showed higher basal glycolysis (*Figure 3E and F*), basal PER (*Figure 3G*) and higher compensatory glycolysis (*Figure 3H*) compared to the FB group. Culture under atmospheric oxygen (IVF20% group) resulted in significantly increase in glycolysis. Overall, these results show that IVF-generated embryos favor glycolysis over oxidative phosphorylation (*Figure 3I and J*), indicating an accentuation of Warburg metabolism.

### IVF-generated embryos show alterations of Intracellular/extracellular pH, NAD, lactate, and pyruvate levels

To gain a more detailed understanding of metabolic alterations occurring in IVF embryos we measured pH and selected metabolites levels. IVF-generated blastocysts had significantly higher intracellular pH (*Figure 4A and B*), while extracellular pH level was lower and showed an inverse correlation to intracellular pH (*Figure 4C*).

Nicotinamide adenine dinucleotide (NAD) levels were significantly lower in IVF-generated embryos (*Figure 4D*) compared to FB group.

Next, we measured intracellular pyruvate and lactate levels, the final metabolites of glycolysis.

Intracellular pyruvate levels were decreased (*Figure 4E*), while intracellular lactate levels were increased (*Figure 4F*) in IVF-generated embryos compared to in vivo embryos.

Extracellular lactate and pyruvate levels were decreased in culture media of IVF embryos (*Figure 4—figure supplement 1A,B*).

### IVF-generated embryos show alteration in enzymes involved in lactate metabolism

Given the critical role of lactate in metabolism (*Brooks, 2020*) we performed both an unsupervised analysis of the blastocyst proteome and analyzed key enzymes involved in lactate metabolism. Proteomic analysis revealed that multiple proteins involved in glycolysis (n=5 proteins), pentose phosphate pathway (PPP, n=3 proteins) and tricarboxylic cycle (TCA, n=6 proteins) were altered in

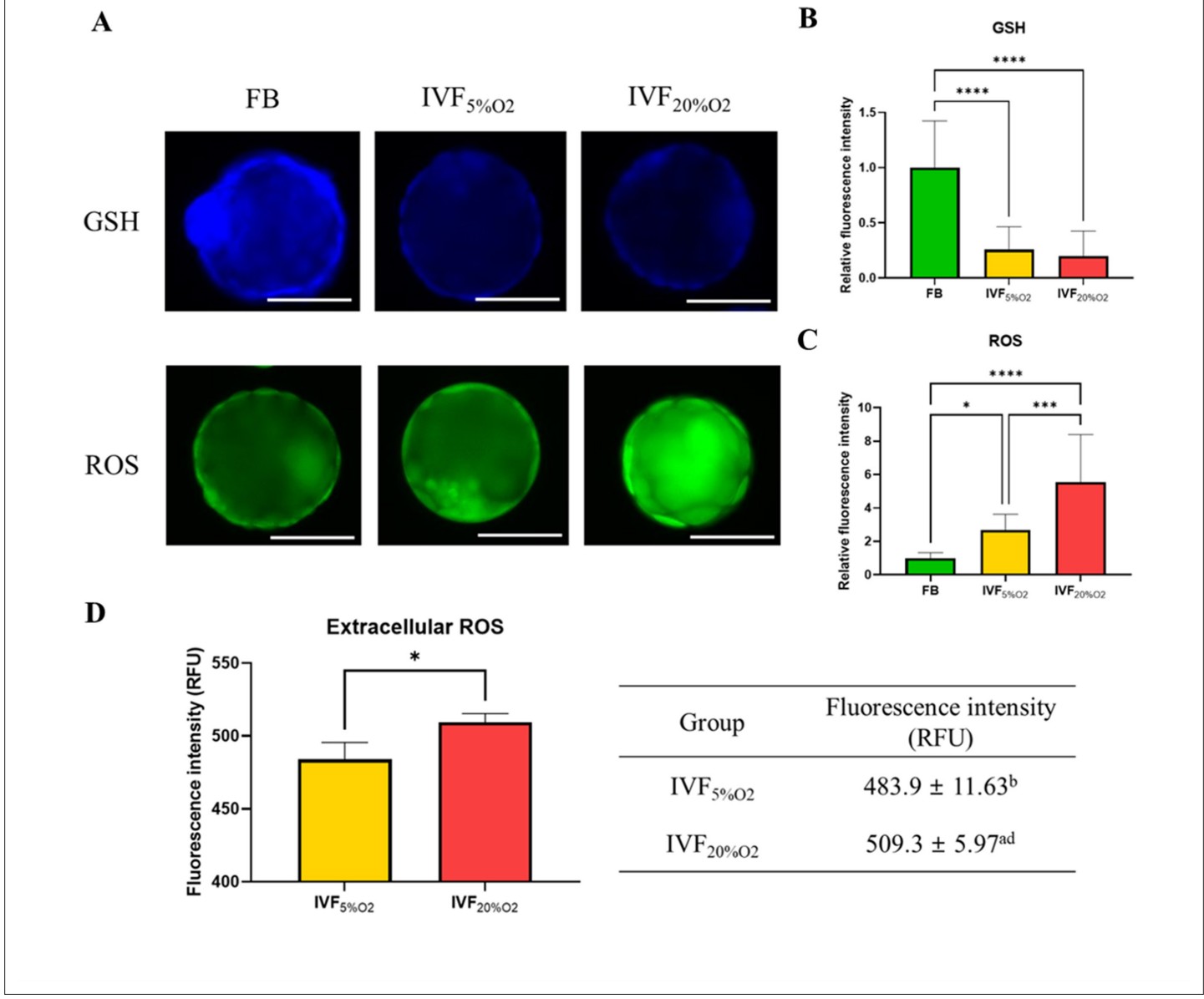

**Figure 1.** The levels of glutathione (GSH) and reactive oxygen species (ROS).

(**A**) Analysis of glutathione (GSH) and reactive oxygen species (ROS) in blastocysts. (**B**) GSH levels are lower in in vitro fertilization-generated blastocysts, while ROS levels are higher both (**C**) intracellularly and (**D**) extracellularly. Since flushed blastocyst (FB) were not cultured, the ROS value for the FB group is missing. Data are shown as means ± SD and at least three independent replicates were performed. Error bar indicates standard deviation. One asterisk (*) if p<0.05. Three and four asterisks (*** and ****) if p<0.001 and<0.0001, respectively. Bar = 50 µm.

The online version of this article includes the following source data and figure supplement(s) for figure 1:

**Source data 1.** The ROS and GSH level in embryos (*Figure 1A–C*).

**Source data 2.** The extracellular ROS level in embryo culture medium (*Figure 1D*).

**Figure supplement 1.** Morphology of blastocyst generated after natural mating (in vivo or flushed blastocyst-group, (**A**) or by in vitro fertilization (IVF) using physiologic oxygen (**B**) IVF5%O$_2$ group, or IVF20%O$_2$ (**C**) IVF 20%O$_2$ group).

**Figure supplement 1—source data 1.** The cleavage and blastocyst rate in embryos (*Figure 1D-E*).

**Figure supplement 1—source data 2.** The total cell number of blastocysts (*Figure 1F*).

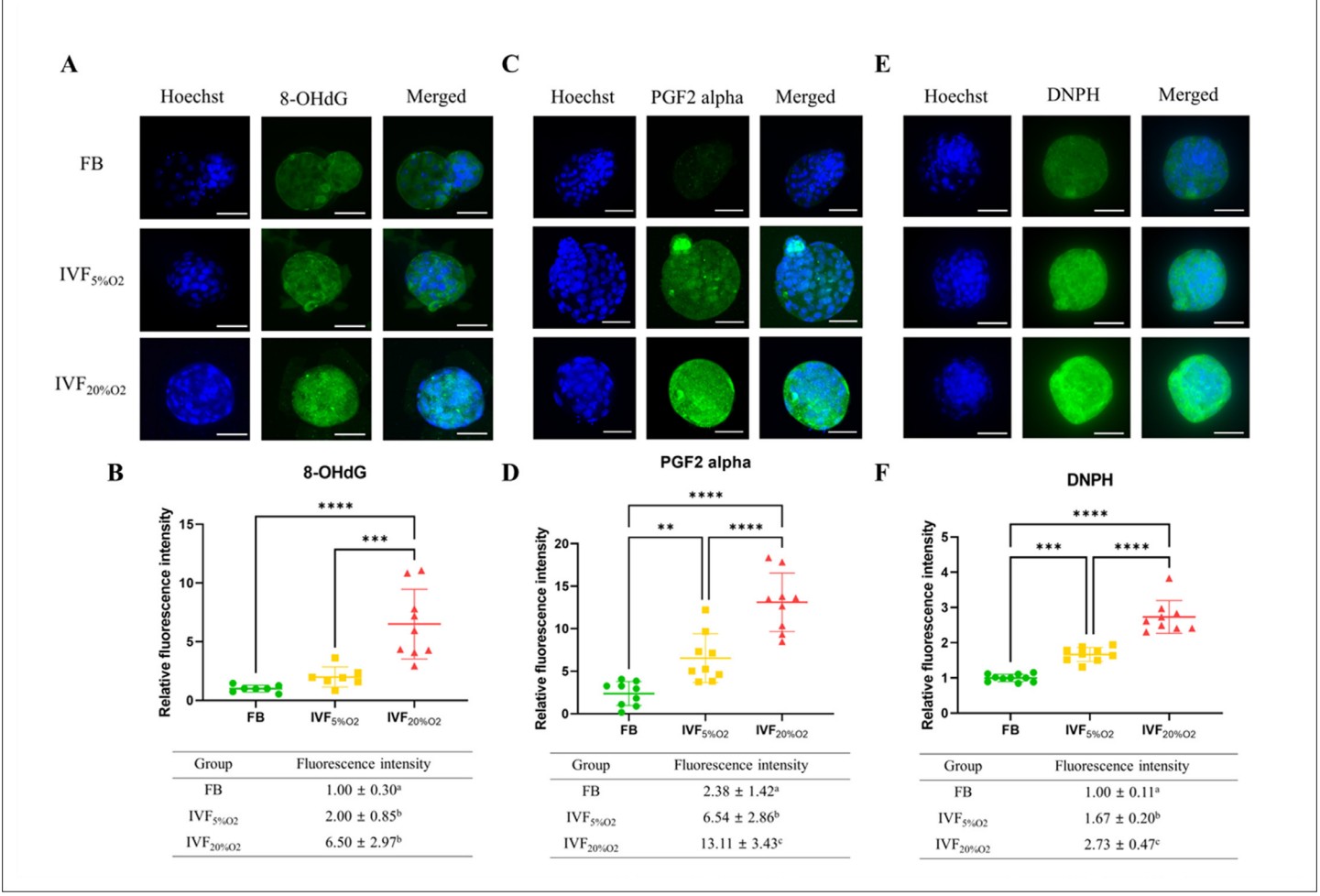

**Figure 2.** Immunocytochemical analysis of oxidative-related markers in blastocysts. (**A and B**) In vitro fertilization (IVF) generated blastocysts show increase in 8-OHdG staining (DNA damage marker, green), (**C and D**) PGF2-alpha (lipid damage marker, green) and (**E and F**) 2,4-dinitrophenylhydrazine (DNPH) staining (protein damage marker, green) compared to in vivo control embryos. Data are shown as means ± SD and at least three independent replicates were performed. Error bar indicates standard deviation. * if $p<0.05$; ** if $p<0.01$, *** if $p<0.001$ and **** if $p<0.0001$. Bar = 50 μm.

The online version of this article includes the following source data for figure 2:

**Source data 1.** The level of oxidative damage (DNA, lipids, and proteins) in embryos (*Figure 2A–F*).

IVF-generated embryos (*Figure 5*). Regarding glycolytic proteins, hexokinase 2, and triosephosphate isomerase 1 were upregulated, while 2,3-bisphosphoglycerate mutase, pyruvate kinase, and LDH-B were downregulated in both IVF groups compared to in vivo generated blastocysts. In addition, LDH-A isoenzyme that favors conversion of pyruvate to lactate *Markert et al., 1975* was upregulated only in IVF5%O₂ blastocysts compared to control.

To gain a further understanding of the etiology of altered lactate levels in IVF-generated embryos, we further measured lactate dehydrogenase (LDH-A and LDH-B) and the level of monocarboxylate transporter 1 (MCT1) protein (involved in transport of lactate through the plasma and inner mitochondrial membrane) by immunofluorescence (*Figure 6*). Both LDH isoenzymes were reduced in IVF-generated embryos (*Figure 6A–C*) compared to in vivo embryos. Similarly, the levels of the transmembrane lactate transporter MCT1 were decreased in IVF embryos compared with FB group (*Figure 6D and E*).

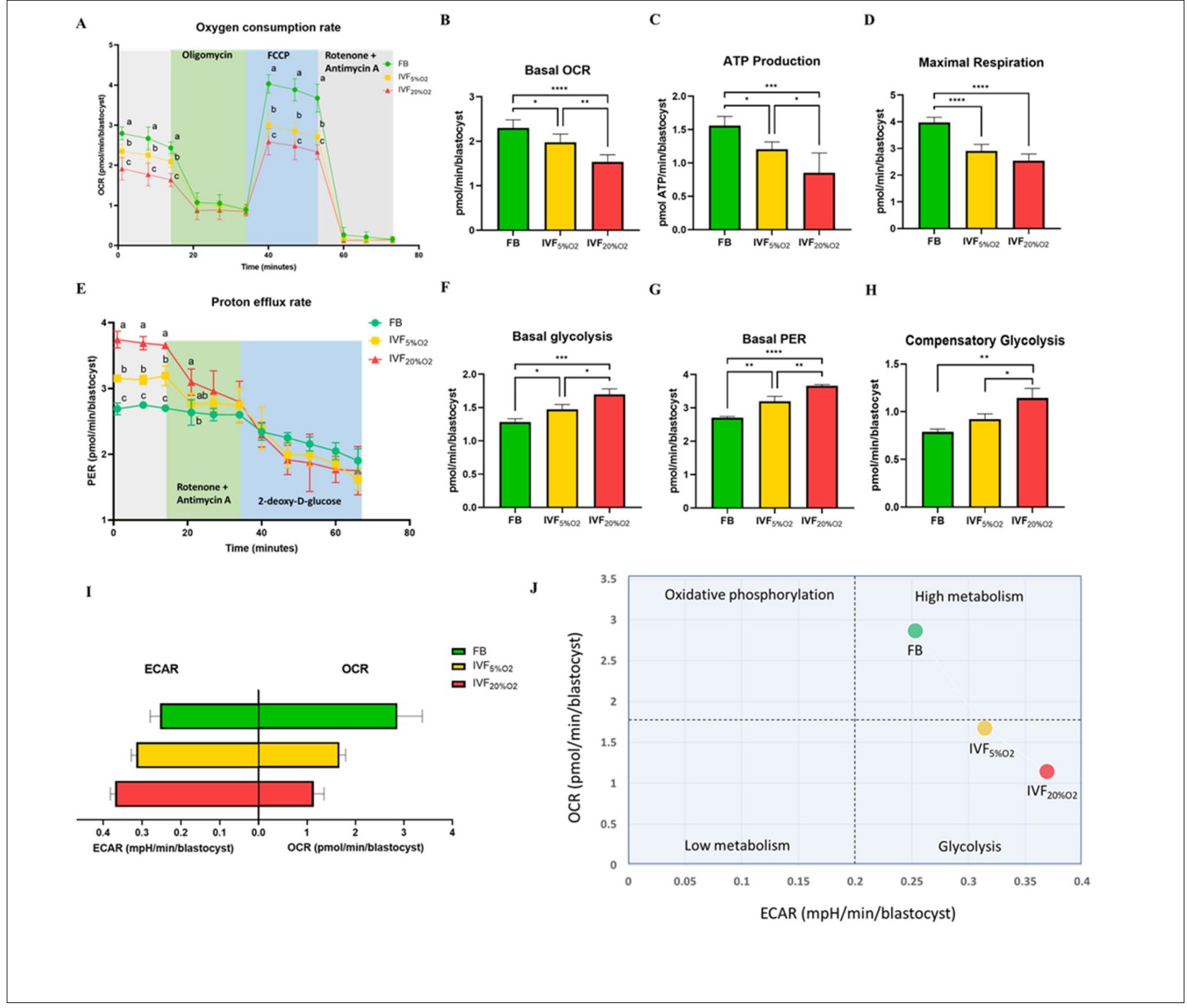

**Figure 3.** Metabolic analysis of mitochondrial and glycolytic function using Agilent Seahorse XF HS Mini. (**A**) Mitochondrial respiration was analyzed at baseline or after injection of oligomycin (Adenosine triphosphate [ATP] synthase inhibitor), carbonyl cyanide-p-trifluoromethoxyphenylhydrazone (FCCP, an uncoupler of oxidative phosphorylation), and rotenone/antimycin A (Complex I/III inhibitors). (**B**) In vitro fertilization (IVF)-generated blastocysts show reduced basal oxygen consumption rate (flushed blastocyst [FB]: 2.3 ± 0.2 pmol/min/blastocyst, IVF5%O$_2$: 2.0 ± 0.2 pmol/min/blastocyst, IVF20%O$_2$: 1.5 ± 0.2 pmol/min/blastocyst), (**C**) ATP production (FB: 1.6 ± 0.1 pmol/min/blastocyst, IVF5%O$_2$: 1.2 ± 0.1 pmol/min/blastocyst, IVF20%O$_2$: 0.9 ± 0.3 pmol/min/blastocyst) and (**D**) maximal respiration (FB: 4.0 ± 0.2 pmol/min/blastocyst, IVF5%O$_2$: 2.9 ± 0.2 pmol/min/blastocyst, IVF20%O$_2$: 2.5 ± 0.2 pmol/min/blastocyst) compared to in vivo-generated embryos. (**E**) Glycolysis was studied at baseline or after injection of rotenone/antimycin and 2-deoxy-D-glucose (2-DG, glycolysis inhibitor). (**F**) IVF generated embryos showed increase in basal glycolysis (FB: 1.3 ± 0.0 pmol/min/blastocyst, IVF5%O$_2$: 1.5 ± 0.1 pmol/min/blastocyst, IVF20%O$_2$: 1.7 ± 0.1 pmol/min/blastocyst), (**G**) basal proton efflux rate (FB: 2.7 ± 0.0 pmol/min/blastocyst, IVF5%O$_2$: 3.2 ± 0.2 pmol/min/blastocyst, IVF20%O$_2$: 3.7 ± 0.0 pmol/min/blastocyst) and (**H**) compensatory glycolysis rate (FB: 0.8 ± 0.0 pmol/min/blastocyst, IVF5%O$_2$: 0.9 ± 0.1 pmol/min/blastocyst, IVF20%O$_2$: 1.1 ± 0.1 pmol/min/blastocyst) compared to in vivo generated embryos. Overall, IVF embryos utilize more glycolysis and less oxidative phosphorylation to generate energy. (**I**) Bar chart of blastocysts ranked by oxygen consumption rate (OCR) to extracellular acidification rate (ECAR) ratio along the X-axis. (**J**) Energy maps of blastocysts. Data are shown as means ± SD and at least three independent replicates were performed. Error bar indicates standard deviation. Different superscript indicates statistically significant differences. * if p<0.05; ** if p<0.01, *** if p<0.001 and **** if p<0.0001.

The online version of this article includes the following source data for figure 3:

*Figure 3 continued on next page*

*Figure 3 continued*

**Source data 1.** The mitochondrial activity in embryos (*Figure 3A*).

**Source data 2.** The mitochondrial activity in embryos (*Figure 3B–D*).

**Source data 3.** The glycolytic function in embryos (*Figure 3E–H*).

**Source data 4.** The ratio of oxygen consumption rate and extracellular acidification rate in embryos (*Figure 3I–J*).

## Several tissues of IVF-generated mice show downregulation of LDH-B and MCT1 proteins

We and others (*Donjacour et al., 2014*; *Feuer et al., 2014a*; *Ceelen et al., 2008*) have shown that IVF offspring, compared to naturally conceived controls, manifest glucose intolerance and several metabolic alterations (*Feuer et al., 2014a*). Re-analysis of our past metabolomic data showed a trend for lower lactate in liver (p=0.05), fat (p=0.06), and serum (p=0.08) of IVF5%O$_2$ conceived mice compared to control (*Feuer et al., 2014a*; *Figure 7—figure supplement 1*).

These findings, together with the reduced levels of LDH-B and MCT1 in IVF embryos prompted us to measure the same enzymes in tissue of adult mice. We found that IVF conceived mice have reduced levels of LDHB (that converts lactate into pyruvate) and MCT1 (involved in transport of lactate through the plasma membrane) proteins in gastrocnemius (*Figure 7A–C*), fat tissue (*Figure 7D–F*), and liver (*Figure 7G–I*). These data suggest that alteration in lactate metabolism may represent a hallmark of altered metabolic function in ART offspring.

## Discussion

This study reports numerous effects of IVF and embryo culture that will be discussed sequentially: (1) mouse blastocysts generated by IVF show increased in ROS production, subsequent oxidative damage, and a plethora of metabolic alterations, (2) oxidative to glycolytic metabolic shift owing to decreased mitochondrial functions and compensatory changes in glycolysis, altered expression of lactate shuttle proteins including LDH and MCT1, and (3) changes in cell redox status and pH, and (4) long-term metabolic dysregulation n IVF animals. In IVF-generated embryos. Importantly, when atmospheric oxygen was used to culture embryos, the phenotype further worsened.

While several studies have suggested an increase in ROS and alteration of antioxidant balance in IVF-generated embryos (*Cebral et al., 2007*; *Goto et al., 1993*; *Martín-Romero et al., 2008*), this study also shows increased oxidative damage to embryos. ROS are second messengers with a likely tight therapeutic window. In fact, during embryo development from the 2 cell to the blastocyst stage, the levels of ROS are increased to maintain their metabolic process (*Ryan et al., 1998*). However, excessive ROS levels causes a lower oocyte maturation during in vitro maturation and lower embryonic development (*Johnson and Nasr-Esfahani, 1994*; *Luvoni et al., 1996*) in addition to cellular apoptosis, DNA damage, inhibition of DNA synthesis, lipid peroxidation, and oxidative modification of proteins (*Gardner and Fridovich, 1991*; *Van Blerkom, 2011*).

The fact that IVF-generated embryos show altered Redox imbalance is also shown by their reduced level of glutathione, since GSH is one of the main non-enzymatic defenses against ROS in mammalian embryo (*Rizzo et al., 2012*; *Takahashi et al., 1993*). The combination of increased ROS and reduced GSH is likely responsible for the increased oxidative damage to DNA, lipid and protein observed in IVF embryos.

Of note, the 8-OHdG is a sensitive indicator of DNA damage and high 8-OHdG levels were detected in patients with low fertilization rates and low blastocyst development (*Nishihara et al., 2018*). The 8-epi-prostaglandin F2-α (PGF2α, F2-isoprostane) and 2,4-dinitrophenylhydrazine (DNPH) are widely regarded as specific and stable oxidative stress indicators to measure the degree of lipid peroxidation and protein oxidation, respectively (*Gongadashetti et al., 2021*; *Halliwell and Lee, 2010*; *Keller et al., 1993*; *McKinney et al., 2000*; *Montuschi et al., 2003*; *Psathakis et al., 2004*).

The second important finding of this paper is the alteration in mitochondria and glycolytic function of IVF-generated embryos. That IVF embryos have reduced OCR rate is not surprising, given that we *Belli et al., 2019* and others *Marei et al., 2019* have shown alteration in mitochondria morphology and mitochondrial membrane potential. The current analysis further adds to the knowledge by showing that IVF embryos have reduced ATP production via mitochondrial respiration and

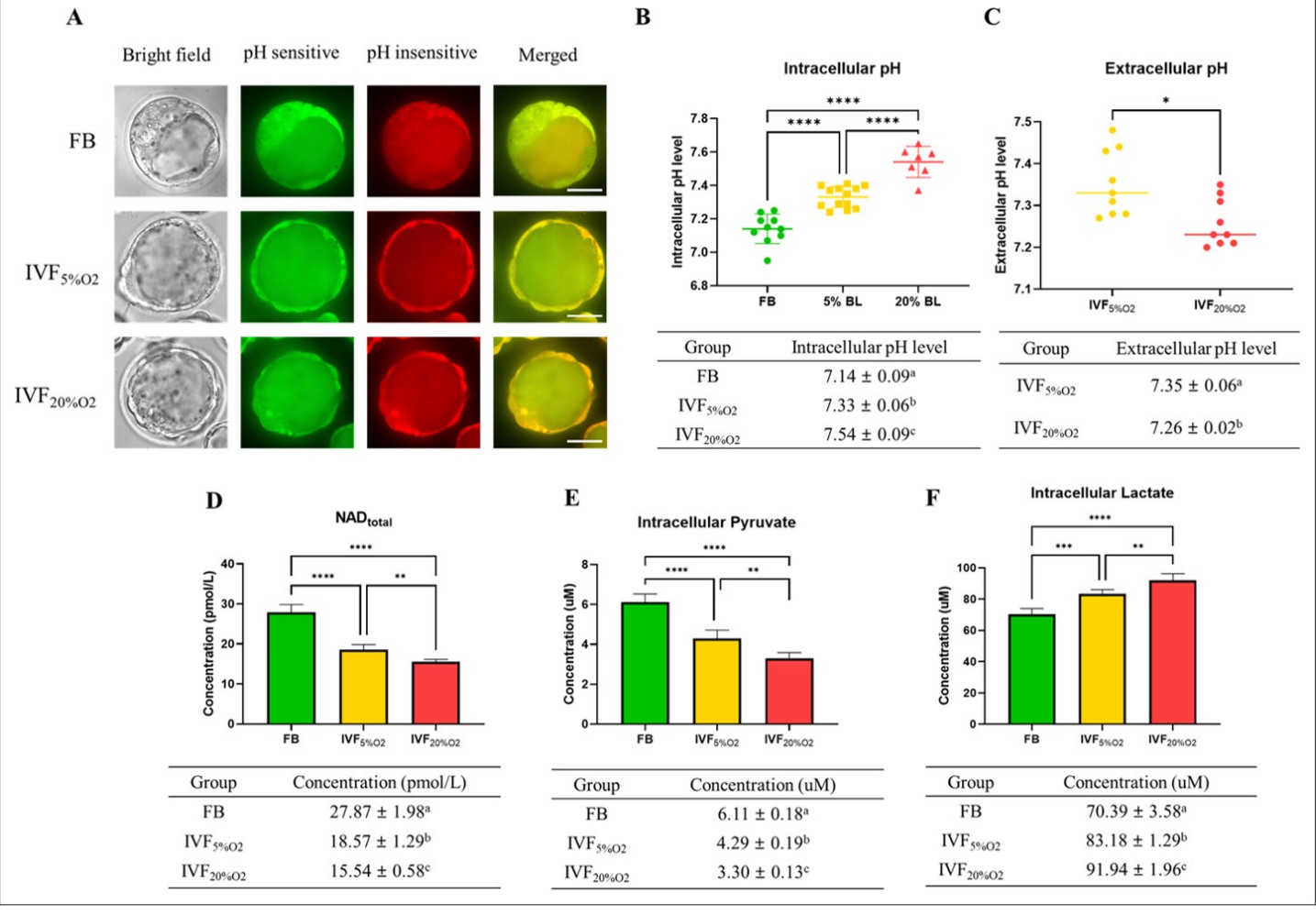

**Figure 4.** IVF-generated embryos show several metabolic alterations. (**A**) Confocal microscopy images of blastocysts using phase contrast or confocal microscopy. Two fluorescence emission wavelengths were used, 640 nm (pH sensitive, green) and 600 nm (pH insensitive, yellow) with an excitation wavelength of 535 nm. (**B**) In vitro fertilization (IVF)-generated embryos show higher intracellular pH and (**C**) lower extracellular pH in culture medium. Of note extracellular pH levels of in vivo-generated embryos are missing since in vivo embryos are not cultured. (**D**) IVF-generated embryos also showed reduction in nicotinamide adenine dinucleotide (NAD) levels and (**E**) intracellular pyruvate levels, but higher intracellular lactate (**F**). Values are mean ± SD. Data are shown as the means ± SD and at least three independent replicates were performed except pyruvate and lactate assay. For pyruvate and lactate assay, total 300 blastocysts were used for one biological replication and seven technical replications. Error bar indicates standard deviation. * if p<0.05; ** if p<0.01, *** if p<0.001 and **** if p<0.0001. Bar = 50 μm.

The online version of this article includes the following source data and figure supplement(s) for figure 4:

**Source data 1.** Intracellular pH level in embryos (*Figure 4A–B*).

**Source data 2.** The NAD level in embryos (*Figure 4D*).

**Source data 3.** The intracellular pyruvate level in embryos (*Figure 4E*).

**Source data 4.** The intracellular lactate level in embryos (*Figure 4F*).

**Figure supplement 1.** Lactate (**A**) and pyruvate (**B**) levels are lower in media where in vitro fertilization (IVF)-generated embryos are cultured compared to media maintained in incubator without embryos.

**Figure supplement 1—source data 1.** The extracellular lactate and pyruvate level in embryo culture medium (*Figure 4—figure supplement 1A-B*).

show a compensatory increase in glycolysis (*Figure 3*). The increase is particularly evident in the IVF20%O$_2$ embryos, that appear to further need to generate energy via glycolysis even when supra-physiologic oxygen is present. Previous studies have shown that the glycolytic rate (percentage of glucose converted to lactate) for mouse blastocysts flushed from the uterus is less than 40%, whereas it increases to over 75% after 3 hr of in-vitro culture. This is important, since higher glycolytic rate was associated with lower implantation rates (*Lane and Gardner, 1996*; *Lane and Gardner, 1998*) and

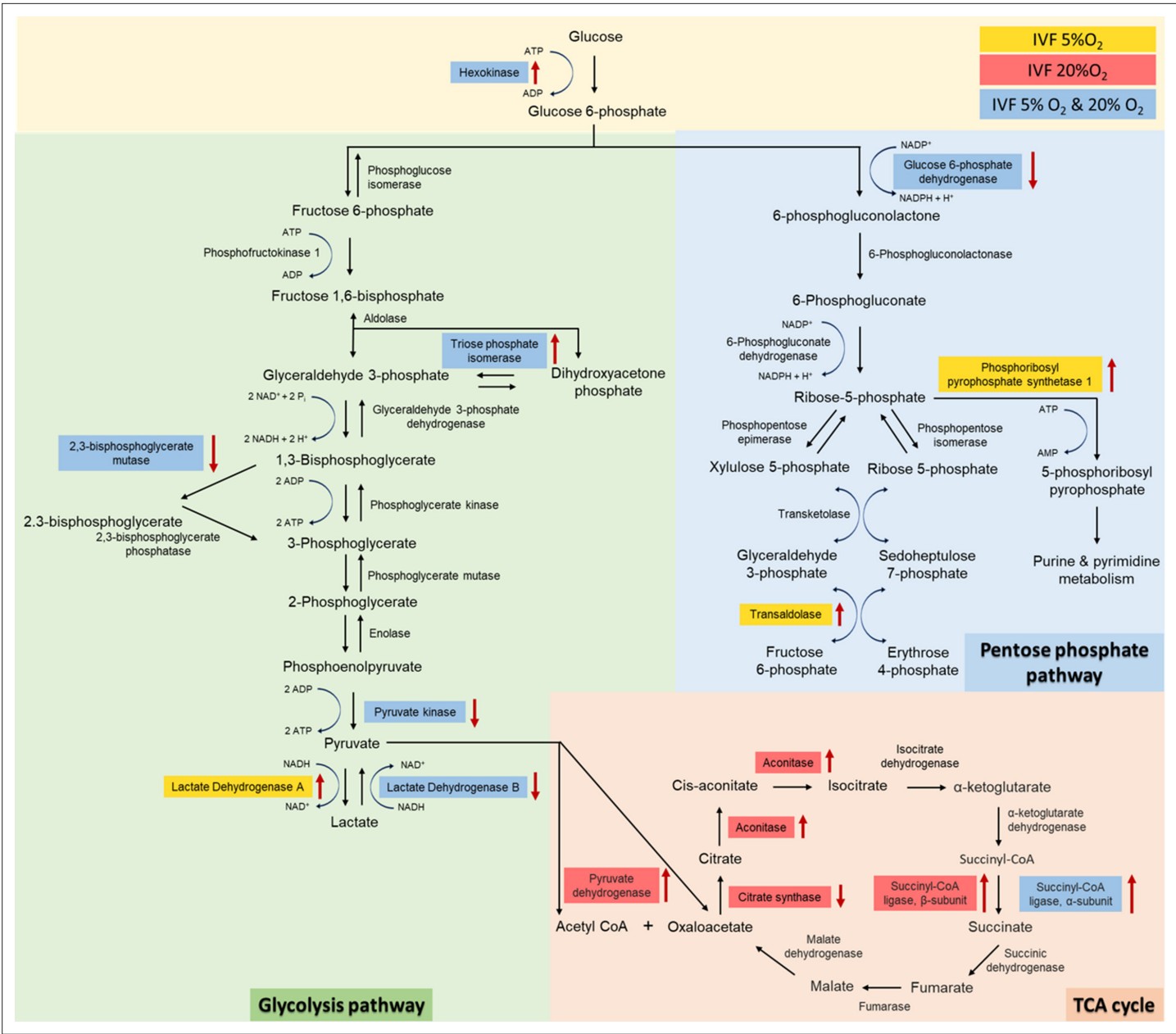

**Figure 5.** Analysis of glycolysis, pentose phosphate pathway (PPP) and tricarboxylic acid (TCA) cycle proteins dysregulated in in vitro fertilization (IVF)-generated blastocysts vs control. Overall, 3965 proteins were identified and 1029 were differentially expressed between the groups. IVF5%O$_2$-generated blastocysts had 334 proteins upregulated relative to the control (flushed blastocyst), while 499 proteins were upregulated on IVF20%O$_2$ embryos, with 85% overlap. Six glycolytic proteins had different levels in IVF embryos compared to control. Hexokinase 2 (HK) and triosephosphate isomerase 1 (TPI) were upregulated, while 2,3-bisphosphoglycerate mutase (BPGM), pyruvate kinase (PK), and lactate dehydrogenase B (LDHB) were downregulated in both IVF groups compared to in vivo generated blastocysts. In addition, lactate dehydrogenase A (LDHA) isoenzyme was upregulated only in IVF5%O$_2$ blastocysts compared to control. Three proteins involved in the pentose phosphate pathway were different between IVF embryos and control embryos. Glucose 6-phopshate dehydrogenase (G6PD) was decreased in both IVF groups, while phosphoribosyl pyrophosphate synthetase 1 (PRPS1) and transaldolase 1 (TALDO1) were higher in IVF5%O$_2$-generated blastocysts compared to control. Four proteins associated with TCA cycle were significantly dysregulated in IVF-generated embryos. Succinyl-CoA ligase α (SUCLG1) was significantly upregulated in both IVF group compared to in vivo embryos. The remaining proteins were dysregulated only in IVF20%O$_2$–generated embryos compared to control: pyruvate dehydrogenase (PDHA1), aconitase (ACO$_2$), and succinyl-CoA ligase β (SUCLG2) were upregulated, while citrate synthase (CS) was downregulated. Proteins are highlighted in blue color, if they were statistically different (p<0.05) in both IVF groups compared to control; yellow color indicates changes of only IVF5%O$_2$-generated blastocysts vs in vivo embryo; red color indicates changes of only IVF20%O$_2$-generated blastocysts vs in vivo embryo; black arrows represent the directionality of the interaction (increased or decreased).

*Figure 5 continued on next page*

*Figure 5 continued*

The online version of this article includes the following source data for figure 5:

**Source data 1.** The proteomic analysis data (*Figure 5*).

indeed our past experiments have shown that mouse embryos generated by IVF have lower live birth rate compared to embryos generated in vivo (*Xiao et al., 2020*).

Hence, it appears that vitro-derived blastocysts increase glycolysis to overcome their impaired mitochondrial function. In future observations, linking glucose consumption, lactate ion and proton efflux rates to embryo viability may be useful as predictors of human embryo viability in ART laboratories.

Among the additional metabolic alterations found in IVF-generated embryos, the reduction in NAD levels and increase in intracellular pH are very relevant.

NAD is an essential coenzyme and an obligate cofactor for the catabolism of metabolic fuels in all cell types (*Frederick et al., 2016*). Growing evidence has confirmed that NAD controls diverse cell signaling pathways (*Yamaguchi and Yoshino, 2017*; *Yang and Sauve, 2016*). Importantly, age-related

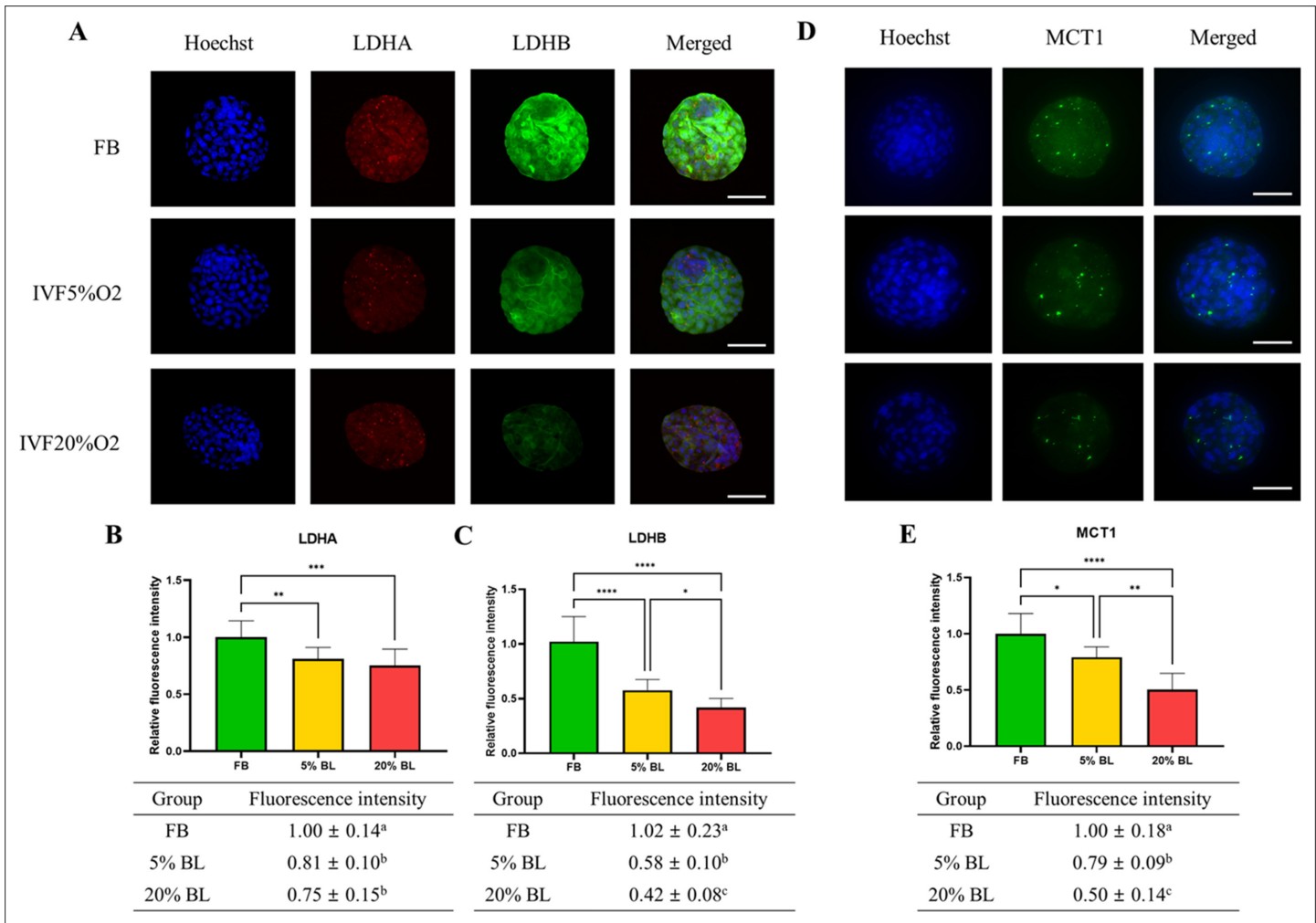

**Figure 6.** IVF-generated embryos show downregulation of enzymes involved in lactate metabolism. (**A**) Expression levels of lactate dehydrogenase (LDH)-A and LDH-B (red color: LDHA, green color: LDHB) and (**B and C**) their quantification. (**D and E**) MCT1 enzyme is downregulated in in vitro fertilization (IVF) embryos compared to control. The expression level of LDHB and MCT1 was decreased in IVF20%O$_2$ compared with IVF5%O$_2$ blastocysts. Data are presented as means ± SD and at least three independent replicates were performed. Error bar indicates standard deviation. * if p<0.05; ** if p<0.01, *** if p<0.001 and **** if p<0.0001. Bar = 50 µm.

The online version of this article includes the following source data for figure 6:

**Source data 1.** The fluorescence intensity of LDHB, LDHA, and MCT1 in embryos (*Figure 6A–E*).

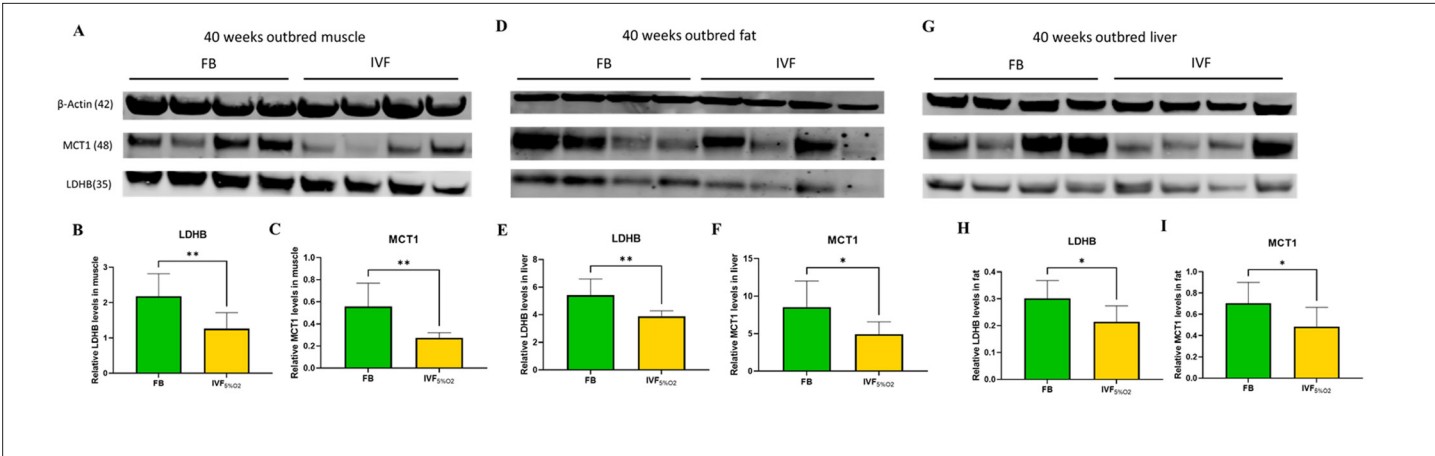

**Figure 7.** Western blot analysis of protein levels of lactate dehydrogenase (LDH)-B and MCT1 enzymes in selected insulin sensitive tissues of 40-week-old mice conceived in vivo (FB n=8, 50% females) or by IVF5%O₂ (n=8, 50% females). In vitro fertilization (IVF)-conceived mice show reduced levels of LDHB and monocarboxylate transporter 1 (MCT1) in gastrocnemius muscle (**A–C**) gonadal adipose tissue (**D–F**) and liver (**G–I**). Selected blots are shown. Data are presented as means ± SD and at least three independent replicates were performed. Error bar indicates standard deviation. * if p<0.05; ** if p<0.01. FB: flushed blastocyst.

The online version of this article includes the following source data and figure supplement(s) for figure 7:

**Source data 1.** Eight biological replicates for FB and IVF were compared.

**Figure supplement 1.** Unsupervised metabolomic analysis by LC/MS was performed in liver, gonadal adipose tissue, and serum of in vivo conceived mice or mice conceived by in vitro fertilization (IVF) (5%O₂).

**Figure supplement 1—source data 1.** The lactate level in liver, fat, and serum in adult mice.

reductions in NAD concentration were linked to declining oocyte quality and infertility, and restoration of NAD levels is advocated for the treatment of age-related infertility (*Bertoldo et al., 2020*). The findings of reduced NAD levels in IVF-derived embryos confirm their significant alteration in energy metabolism and open the possibility that culture media supplemented with NAD might help to overcome their impaired energy metabolism.

The increase in intracellular pH and decreased in extracellular pH found in IVF-generated embryos is particularly important, because it confirms the profound alteration in embryo metabolism. In fact, cancer cells are known to have higher intracellular pH and lower extracellular pH (*Webb et al., 2011*) and regulation of intra- and extracellular pH is considered essential for cellular physiological and metabolic function (*Becker and Deitmer, 2021*).

Monocarboxylate transporter imbedded in plasma and other cell membranes are symporters moving monocarboxylate anions (e.g. lactate) and protons in a stoichiometric (1:1) ratio. Not surprisingly there is a well-known correlation between intracellular and extracellular pH: while extracellular pH affects intracellular pH (*Edwards et al., 1998b*), if the embryos produces excess of metabolic acids (either in the form of lactate, protons, or $CO_2$), extracellular acidification occurs (*Gatenby and Gillies, 2008*; *Gillies et al., 2012*; *Parks et al., 2011*). The existence of this correlation is particularly valuable because precise measurement of extracellular pH could provide insight into the embryonic metabolism and offer diagnostic opportunities to identify healthy embryos. Given that individual blastocysts show variation in intracellular pH, oxidative damage, ROS, NAD levels, and based on the Goldislocks principle of embryo health (*Leese et al., 2016*), embryo showing less stress could be identified and selected for early transfer.

We next found that IVF-generated embryos showed decrease in pyruvate and increase in lactate levels. Pyruvate and lactate are key energy sources for the preimplantation embryo. Pyruvate is the preferred energy substrate during the cleavage stage, while glucose consumption is low but tends to increase at the blastocyst stage, when also oxidative phosphorylation increases (*Leese et al., 2016*). Importantly, while lactate was once believed to be a waste product of anaerobic metabolism, it is now clear that it plays a key role in physiology and metabolism (*Brooks, 2018*). Lactate is continuously produced under aerobic conditions and plays 3 major roles: (1) it is a major energy source, (2) the most important gluconeogenic precursor, and (3) functions as a signaling molecule. Notably, the product

of glycolysis is lactate, not pyruvate (*Rogatzki et al., 2015*) and lactate oxidation is dominant in vivo. For example, muscle lactate concentration exceeds that of pyruvate by one (10 ×) to two (200–400 ×) orders of magnitude in resting and exercising human muscles, respectively (*Henderson et al., 2004*). Our findings confirm that lactate concentration is tenfold higher than the lactate concentration, but IVF embryos show a further increase in lactate production since the lactate to pyruvate ratio is 11 for FB embryos, 19 for IVF5%$O_2$, and 27 for IVF20%$O_2$. We can therefore hypothesize that the increased level of intracellular lactate in IVF embryos might be a strategy adopted by the embryo to survive the suboptimal culture environment.

The next studies aimed to clarify the molecular mechanism leading to increase lactate levels in IVF embryos. Unsupervised global proteomics analysis revealed that LDH-B was downregulated in IVF embryos. We confirmed these results by performing immunofluorescence studies and found that IVF embryos showed downregulation of both LDHA and B and of the monocarboxylate transporter, MCT1, providing an explanation for the increase in their lactate levels. LDH is a tetramer composed of two subunits (LDH-M, found in muscles and LDH-H, found in heart), encoded by the LDHA and LDHB genes, that catalyzes the reversible conversion of pyruvate to lactate using NAD as a cofactor (*Markert et al., 1975*). Lactate production is associated with increase in $NAD^+$ (*Shibata et al., 2021*). Importantly, although different isozymes possess different activities (*Summermatter et al., 2013*), the reduction in protein levels of both isozymes in IVF embryos suggest that LDH is a critical nodus or regulation.

Both LDH-A and B isoenzymes are active in the preimplantation mouse embryos, but LDH-B is the principal form. LDH-A will become predominant at the time of implantation (*Auerbach and Brinster, 1967*; *Lane and Gardner, 2000*). Human studies suggest a similar pattern: the mRNA of LDH-B was detected in human oocytes, 4- and 8-cell embryos, while the mRNA for LDH-A was detected only in two of four oocytes and one of three 8-cell embryos. However, the mRNA for testis-specific LDH-C was not detected (*Li et al., 2006*).

MCT1 is the major transporter for lactate and belongs to a family of proteins involved in the transport of monocarboxylates, such as lactate, pyruvate, and ketone bodies, across the plasma membranes. MCT1 is a proton-linked transporter, i.e., it ensures symport of monocarboxylates and protons in a 1: 1 ratio (*Chatel et al., 2017*). Its downregulation further explains the profound alteration in lactate metabolism in IVF embryos.

Finally, we found that tissues of adult mice conceived by IVF also showed reduced levels of LDH-B and MCT1 proteins in gastrocnemius, fat tissue, and liver and had a trend for lower lactate levels (*Figure 7—figure supplement 1*). The dysregulation of LDH-B and MCT1 in both embryo and tissues of adults conceived by IVF strongly suggest an epigenetic regulation. Indeed, embryo culture is associated with profound epigenetic changes (*Doherty et al., 2000*; *Reik, 2007*; *Reik et al., 2001*; *Ruggeri et al., 2020*).

In this work, we have focused on studying the blastocyst stage, since this is the stage at which human embryos are most commonly transferred. Future studies should examine how metabolic differences, ROS and oxidative damage occur during earlier stages of embryo development, both in mice and possibly human embryos. Although ROS production was documented after culture of embryos using commercially available media even under conditions of low oxygen concentration (*Martín-Romero et al., 2008*), design of optimized culture media, aimed at reduce ROS generation should be encouraged. In addition, a possible limitation of the paper has been the prevalent utilization of fluorescence-based microscopy as a method to identify differences in expression of oxidative marks or protein levels.

In summary, we can formulate a hypothesis in which (*Figure 8*) oxidative stress from in vitro condition increase ROS and induce oxidative damage resulting in a shift toward Warburg metabolism, given that lactate is a critical energy source (*Brooks, 2018*). The higher intracellular lactate levels will likely induce epigenetic changes, to favor Warburg metabolism during development, as an embryonic attempt to optimize growth based on the environment predicted to be experienced in the future. A recent study (*Yang et al., 2021*) showed that hypoxic embryo culture alters histone lactylation marks (H3K23la, H3K18la), providing partial support to this hypothesis.

When the environment does not match the prediction, disease risk increases (*Godfrey et al., 2007*). Low lactate would be beneficial in a setting of low food resources because it could favor lipolysis (*Brooks, 2020*). In fact, lactate activates the hydroxycarboxylic acid receptor 1 (HCAR1), a

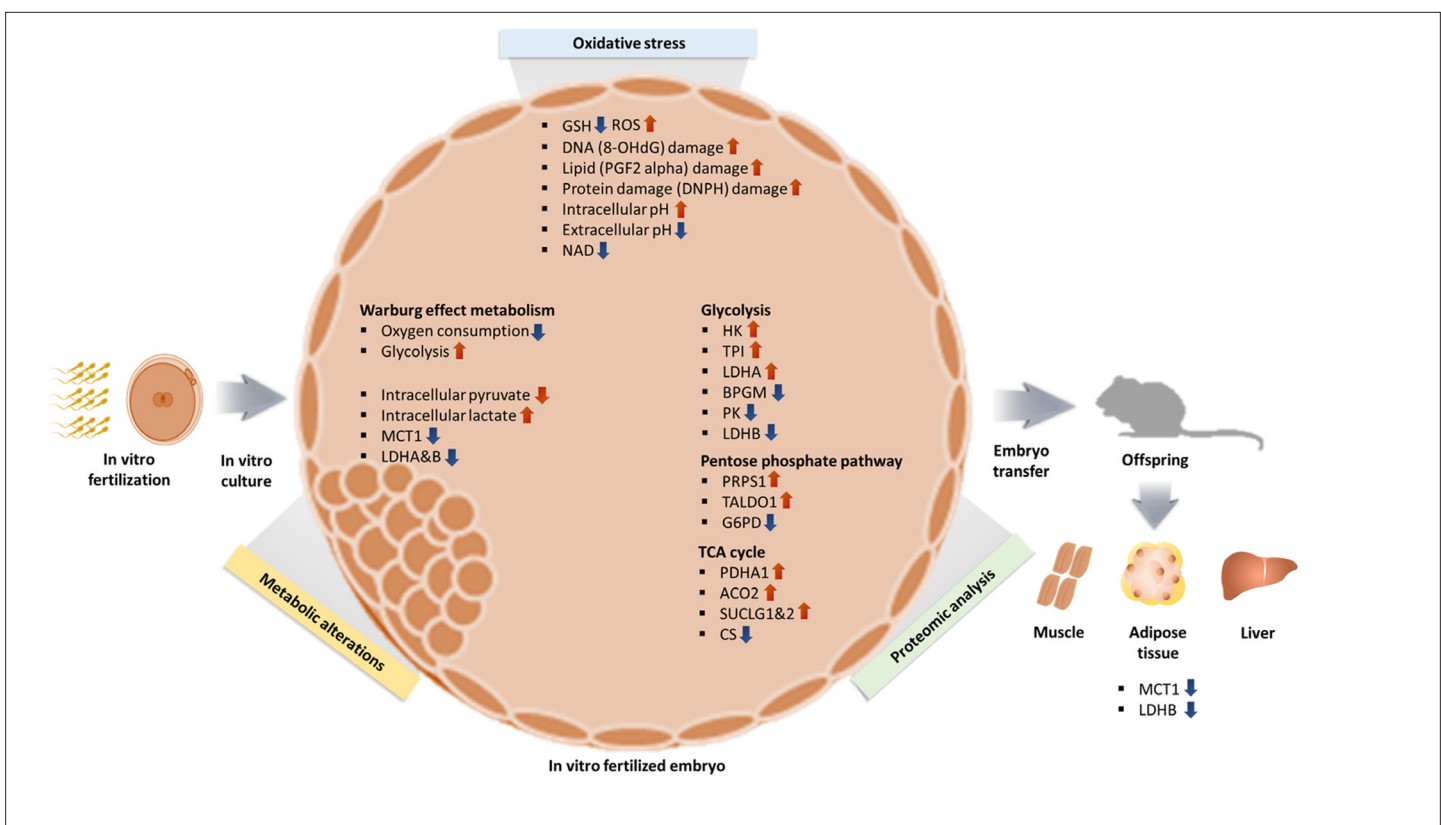

**Figure 8.** Summary of changes observed in IVF generated blastocysts and offspring. In vitro fertilization (IVF) generated embryos show increase in ROS and oxidative damage and significant metabolic derangement including decreased NAD levels, increased intracellular pH and contemporaneous decrease in extracellular pH and increased Warburg metabolism. Ultimately, IVF-generated embryos show increased intracellular lactate levels, likely secondary to decreased level of MCT1, LDH-A, and LDH-B enzymes. Of note, adult mice conceived by IVF also show decreased levels of MCT1 and LDH-B.

G protein-coupled receptor, which in turn inhibits lipolysis in fat cells via cAMP and CREB (*Liu et al., 2009*). However, since there is an abundance of food in our society, this mismatch could predispose IVF concepti to develop chronic disease like glucose intolerance.

## Materials and methods

**Key resources table**

| Reagent type (species) or resource | Designation | Source or reference | Identifiers | Additional information |
|---|---|---|---|---|
| Antibody | Anti-8-Hydroxy-2-deoxyguanosine (Mouse monoclonal) | Abcam | Cat#: ab48508 | IF (1:200) |
| Antibody | Anti-8 iso Prostaglandin F2 alpha (Rabbit polyclonal) | Abcam | Cat#: ab2280 | IF (1:200) |
| Antibody | Anti-DNP antibody (Rabbit polyclonal) | Sigma | Cat#: D9656 | IF (1:200) |
| Antibody | Anti-MCT1 antibody (Mouse monoclonal) | Santa Cruz | Cat#: sc365501 | IF (1:200) |
| Antibody | Anti-LDHA antibody (Rabbit monoclonal) | Cell Signaling | Cat#: 3582 | IF (1:200) |

*Continued on next page*

*Continued*

| Reagent type (species) or resource | Designation | Source or reference | Identifiers | Additional information |
|---|---|---|---|---|
| Antibody | Anti-LDHB antibody (Mouse monoclonal) | Santa Cruz | Cat#: sc100775 | IF (1:200) |
| Antibody | Goat anti-Rabbit (goat polyclonal, Alexa Fluor conjugate) | Abcam | Cat#: ab150077 | IF (1:200) |
| Antibody | Donkey anti-Rabbit (donkey polyclonal, Alexa Fluor conjugate) | Abcam | Cat#: ab150075 | IF (1:200) |
| Antibody | Goat anti-Mouse (goat polyclonal, Alexa Fluor conjugate) | Abcam | Cat#: ab150113 | IF (1:200) |
| Biological sample (mouse) | Blastocyst | This paper | | Female |
| Biological sample (mouse) | Liver | This paper | | Male/Female |
| Biological sample (mouse) | Muscle | This paper | | Male/Female |
| Biological sample (mouse) | Fat | This paper | | Male/Female |
| Commercial assay or kit | Seahorse XF Cell Mito Stress Test kit | Agilent | 103010–100 | |
| Commercial assay or kit | Seahorse XF Glycolysis Stress Test kit | Agilent | 103020–100 | |
| Commercial assay or kit | Pyruvate assay kit | Cayman | Cat#: 700470 | |
| Commercial assay or kit | Lactate assay kit | Cayman | Cat#: 700510 | |
| Commercial assay or kit | NAD Quantitation kit | Sigma | Cat#: MAK037 | |
| Chemical compound, drug | SNARF-1 | Thermo fisher | Cat#: C1272 | |
| Software, algorithm | GraphPad Prism | GraphPad | | |
| Strain, strain background (mouse) | CF-1 | ENVIGO | Hsd:NSA | Oocyte donor |
| Strain, strain background (mouse) | B6D2F1 | The Jackson Laboratory | 100006 | Sperm donor |

## Embryo collection, in vitro fertilization, and embryo culture

Animal experiments were approved by the Institutional Animal Care and Use Committee of the University of California, San Francisco, and all animals were maintained according to the institutional regulation under a 12 hr light/dark cycle with ad libitum access to water and food. In the present study, in vitro fertilization was performed as previously described (*Feuer et al., 2014b*). In brief, five IU pregnant mare serum gonadotrophin (PMSG) was injected to CF1 female mice (8–9 weeks), and after 48 hr later, 5 IU human chorionic gonadotropin (hCG) was injected to those female mice for superovulation. Sperm were collected from the cauda epididymis in B6D2F1 male mice (8–9 weeks) and cumulus-oocyte complexes (COCs) were obtained from ampullae 13–14 hr after hCG administration. The COCs were incubated for 4 hr in human tubal fluid (HTF) medium (Millipore Corp; MR-070-D) with appropriate concentration of sperm obtained following 1 hr capacitation. The fertilized embryos were washed in several drops of KSOM (Millipore, MR-106-D) and cultured in the same medium to the blastocyst stage at 37°C under Ovoil (Vitrolife, #10029). In the present study, zygotes were cultured with different concentration of oxygen until blastocyst stage; (1) 37°C, 5% $CO_2$ in humidified air, 5%

oxygen (IVF 5% group), and (2) 37°C, 5% $CO_2$ in humidified air, 20% oxygen (IVF 20% group). Control embryos (FB) were obtained from flushed blastocysts. Briefly, CF1 female mice (8–9 weeks) were superovulated with administration of 5 IU PMSG and 48 hr later five IU hCG. Then, those females were mated to B6D2F1 males. The vaginal plug was checked after 14–18 hr and this was considered as day 0.5. To control for the known delay in development after culture in vitro, for all experiments, only expanded blastocysts of similar morphology were used, as done before (*Doherty et al., 2000*; *Rinaudo et al., 2006*; *Rinaudo and Schultz, 2004*). The in vivo-generated blastocysts were isolated by flushing 96–98 hr after hCG administration. IVF5%$O_2$ and 20%$O_2$ generated embryos reached the blastocyst stage after 96–98 hr following in vitro culture and 113–114 hr after hCG administration, respectively.

## Evaluation of embryo development and a total cell number of blastocysts

The embryos were cultured for 4 days at 37°C in a humidified atmosphere of different concentration of oxygen (5% and 20%). The evaluation of embryo development including cleavage rate and blastocyst formation rate from IVF and control groups were based on the morphology of the embryos and was approximately as follows, following hCG injection; (1) 2 cell: 16–18 hr after IVF, and (2) blastocyst: 108–110 hr. The blastocysts were stained with 5 µM of Hoechst 33,342 for 7 min to count the total cell number in blastocysts. After washing in the phosphate buffer saline (PBS) medium, stained blastocysts were mounted on a glass slide in a drop of glycerol with gentle compression using a cover slip. Cell counting was performed, and images were obtained under a Nikon scanning confocal microscope fitted with an ultraviolet lamp and 460 nm/560 nm excitation filter.

## Redox status: assessment of GSH/ROS levels in blastocyst

For quantification of intracellular ROS levels, blastocysts were washed twice in poly vinyl alcohol-phosphate buffered saline (PVA-PBS) (1 mg/ml) and incubated in 50 µl droplets of 10 µM 2',7'-dichlorodihydro-fluorescein diacetate (H2DCFDA, D6883, Sigma-Aldrich) in PVA-PBS for 15 min at 37°C in an atmosphere of 5% $CO_2$. Then, the blastocysts were washed three times in PVA-PBS, and transferred to 10 µL droplets of PVA-PBS on a glass slide. Fluorescence intensity for ROS was measured under a Nikon scanning confocal microscope with a filter at 480 nm excitation and 510 nm emission. Evaluation of GSH levels followed same procedure, but the blastocysts were incubated in 10 µM 4-chloromethyl-6,8- difluoro-7-hydroxycoumarin (CMF2HC; Cell Tracker Blue, Life Technologies, Carlsbad, USA). Fluorescence intensity for GSH was measured under a Nikon scanning confocal microscope with a filter at 371 nm excitation and 464 nm emission. The recorded fluorescence intensities were quantified using Image J software (version 1.48; National Institutes of Health, Bethesda, MD, USA) after deducting the background value. Total of 11–17 blastocysts were used in each group for five replications.

## Assessment of ROS concentration in culture medium

ROS concentration in culture was assessed using an Oxiselect in vitro ROS/RNS Assay kit (Cell Biolabs, San Diego, CA, USA). The experiment was processed following the manufacturer's instructions. Briefly, for measuring the total free radical presence in culture medium, the supernatant medium obtained from each group after IVC was obtained. Supernatant were transferred to 1.5 ml tubes and centrifuged at 10,000 g for 5 min to remove insoluble particles. The 50 µl of each supernatant was added to wells of a 96-well plate suitable for fluorescence measurement. Then 50 µl of catalyst was added to each well and incubated for 5 min at room temperature. Last, 100 µl of DCFH solution was added to each well followed by incubation at room temperature for 15 min and reading the fluorescence with a fluorescence plate reader at 480 nm excitation/530 nm emission (Soft Max Pro 4.7.1, Spectra max M2). The ROS concentrations were determined by comparison with the predetermined dichlorodihydrofluorescein (DCF) standard curve.

## Assessment of oxidative stress by immunofluorescence

Immunofluorescence staining was performed to evaluate the degree of oxidative stress in blastocysts derived from each group. Blastocysts were washed three times in PBS containing 0.2% PVA and fixed with 4% paraformaldehyde (w/v) in PBS for 30 min at room temperature. All steps were performed at room temperature unless otherwise stated. After washing three times in PBS, blastocysts were

permeated with 1% (v/v) Triton X-100 in PBS for 2 hr. Then, the samples were washed three times in PBS and blocked with 2% BSA in PBS for 4 hr at 4°C. The blastocysts were incubated with primary antibodies for 8-hydroxy-2-deoxyguanosine (8-OHdG, 1:200; ab48508, Abcam), 8-iso prostaglandin F2 alpha (PGF2 alpha, 1:200; ab2280, Abcam), and 2,4-dinitrophenylhydrazine (DNPH, 1:200; D9656, Sigma), monocarboxylic acid transporter 1 (MCT1, 1:200; sc-365501, Santa Cruz), lactate dehydrogenase A (LDHA, 1:200; #3582, Cell Signaling), lactate dehydrogenase B (LDHB, 1:200; sc-100775, Santa Cruz) at 4°C overnight. Blastocysts were then washed three times in PBS with 2% BSA and then incubated with a secondary anti-rabbit antibody (1:200, ab150077, Abcam; 1:200, ab150075, Abcam) or antimouse antibody (1:200; ab150113, Abcam) for 3 hr at room temperature. After washing three times in PBS with 2% BSA, the samples were mounted on glass slides. Images were captured under a Nikon scanning confocal microscope with the same exposure times and adjustments. The intensities of 8-OHdG, PGF2 alpha, DNPH, MCT1, LDHA, LDHB were measured by Image J software (version 1.46 r; National Institutes of Health). For the intensities of 8-OHdG, PGF2 alpha, DNPH, total of 26–27 blastocysts were used in each group for at least three replications. For the intensities of MCT1, LDHA, LDHB, total of 31–33 blastocysts were used in each group for at least three replications.

## Metabolic profile analyses in blastocysts

Metabolic function in blastocysts were carried out using Agilent Seahorse XF HS Mini Analyzer (Agilent, Santa Clara, CA, USA) according to the manufacturer's instruction. Briefly, for plate hydration, 200 µl of distilled water was added to the sensor-containing Seahorse 8-well fluxpaks (Agilent Technology), which were coated with Poly-D-Lysine, were incubated overnight at 37°C in a non-$CO_2$ humidified incubator. Ten blastocysts were incubated in XF base assay media supplemented with 1 mM glutamine, 0.2 mM glucose, 0.2 mM pyruvate, and incubated at 37°C in a non-$CO_2$ humidified incubator for 1 hr to allow to pre-equilibrate with the assay medium.

For real-time ATP rate analysis, 1.5 µM oligomycin (ATP synthase inhibitor) and 0.5 µM rotenone/ antimycin A (Complex I/III inhibitor) were sequentially injected after incubation. For mitochondrial respiration analysis, 1.5 µM oligomycin, 1 µM Carbonyl cyanide-4-trifluoromethoxyphenylhydrazone (FCCP, a potent uncoupler of oxidative phosphorylation), and 0.5 µM rotenone/antimycin A were sequentially injected. Meanwhile, 0.5 µM rotenone/antimycin A and 50 mM 2-deoxy-D-glucose (2-DG, glycolysis inhibitor) were injected for glycolytic rate analysis. Oligomycin inhibits ATP-synthase and can be considered as indicator of the proportion of $O_2$ consumption directly coupled to ATP generation. FCCP is a potent mitochondrial uncoupler, which dissipates the proton gradient between the intermembrane and the matrix in mitochondria allowing the maximal OCR measurement. As complex I/III inhibitors, rotenone and antimycin A combine to inhibit the electron transport chain entirely; therefore, the proportion of OCR after the addition of rotenone/antimycin A is considered to be non-mitochondrial effect. 2-DG is a glucose analog which inhibits glycolysis through competitive binding of glucose hexokinase in the glycolytic pathway. These measurements were used to determine the magnitudes of various parameters of OCR, ECAR, ATP rate, mitochondrial respiration rate, and glycolytic rate based on the targets of each successive drug injection. ECAR is the sum of two components: respiratory acidification in the form of $CO_2$ (which hydrates to $H_2CO_3$ then dissociates to $HCO_3^- + H^+$) and glycolytic acidification in the form of lactate + $H^+$.

## Measurement of intracellular/extracellular pH in blastocyst and culture medium

Intracellular pH was determined by using the pH-sensitive fluorophore, SNARF-1 (esterified derivative; Molecular Porbes, Eugene, OR), loaded into blastocysts by incubating them with 5 µM SNARF-1 at 37°C for 20 min in KSOM (*Edwards et al., 1998a*). After SNARF-1 staining, blastocysts (7–13 blastocysts per group, in triplicates) were washed with fresh KSOM and placed in a temperature-controlled chamber. Two fluorescence emission wavelengths were detected, 640 nm (pH sensitive) and 600 nm (pH insensitive), using an excitation wavelength of 535 nm. The ratio of the two emission intensities was calculated by dividing the images after background subtraction. At the conclusion of each experiment, the ratio was converted to an actual intracellular pH value using the calibration standard curve, where intracellular pH is clamped to known extracellular values using the ionophores, nigericin (10 µg/ml). The SNARF-1 staining and exposure to excitation illumination does not adversely affect the blastocyst, as previous research demonstrated

that mouse embryos normally cleaved following pH measurements (*Edwards et al., 1998a*). To control for the possible variation in blastocoel size in different embryos, we compared immunofluorescence level of only the inner cell mass and trophoblast region of blastocysts and excluded the blastocoel region.

The extracellular pH level of the culture medium was measured in 30 µl of fluid in which 20 embryos were cultured for 4.5 days by using a pH electrode with a small tip (3 mm in diameter, Orion Ross Electrode, Thermo Scientific, USA) attached to an Accumet basic AB15 pH meter (Mettler Toledo, USA). Briefly, two buffer calibrations (pH 7 and 10) were performed at same temperature as the sample. After calibration, rinse the electrode with distilled water and then place the electrode into the 30 µl of sample. When the reading is stable, pH level was recorded.

## Assessment of nicotinamide adenine dinucleotide (NAD) concentration in blastocysts

Nicotinamide adenine dinucleotide (NAD) concentration was measured using the NAD Quantitation kit (MAK037, Sigma-Aldrich, MO, USA). In brief, NADH (Reduced form of NAD)/NAD was extracted from blastocysts by using NADH/NAD extraction buffer with freeze/thawing for 2 cycles of 20 min on dry ice. The mixture was centrifuged at $13,000 \times g$ for 10 min to remove insoluble material. The NAD/NADH supernatant was transferred into a 1.5 ml Eppendorf tube. The 50 µl of extracted samples were transferred into a 96 well plate. Then, 100 µl of the master reaction mix were added to each well and incubated for 5 min at room temperature to convert NAD to NADH for $NAD_{total}$ determinations. Finally, 10 µl of NADH developer were added into each well and incubated at room temperature for 1–4 hr. The reactions were stopped by adding 10 µl of stop solution into each well and mixing well. The measurement was performed at 450 nm by using microplate reader (Soft Max Pro 4.7.1, Spectra max M2).

## Assessment of pyruvate and lactate level in blastocyst and culture media

As another measure of cytosolix redox state, intracellular and extracellular pyruvate and lactate level were measured in blastocysts and culture medium respectively following manufacturer's instructions (Pyruvate assay kit, Cayman Chemical, MI, USA; Lactate assay kit, Cayman Chemical, MI, USA). To collect 300 blastocysts, we performed multiple IVF, each IVF resulting in 10–20 blastocysts cultured in 30 microliters of media. While intracellular lactate and pyruvate were performed on the embryos collected, the media from different experiments was pooled to a final 500 microliter volume. For sample preparation, the 500 µl of embryo culture medium was collected for measurement of extracellular pyruvate/lactate level. To deproteinate the sample, 500 µl of 0.5 M metaphosphoric acid (MPA) were added and the specimen was placed on ice for 5 min. Then, the mixture was centrifuged at $10,000 \times g$ for 5 min at 4°C to pellet the proteins. The supernatant was removed and the 50 µl of potassium carbonate was added to neutralize the acid. The sample was centrifuged at $10,000 \times g$ for 5 min at 4°C and the supernatant removed for assaying.

For measurement of intracellular pyruvate/lactate levels in blastocysts, 0.5 ml of 0.25 M MPA were added to blastocysts and placed on ice for 5 min. Three-hundred blastocysts per group were used to perform this experiment (900 total). Then, the sample was centrifuged at $10,000 \times g$ for 5 min at 4°C and the supernatant was removed for assaying. Then, 50 µl of assay buffer, 50 µl of cofactor mixture, 10 µl of fluorometric detector, and 20 µl of samples were added to each well. The reactions were initiated by adding 20 µl of enzyme mixture to all of the wells being used. Thereafter, the plate was incubated for 20 min at room temperature and then read using an excitation wavelength between 530–540 nm and emission wavelength between 585–595 nm by using microplate reader (Soft Max Pro 4.7.1, Spectra max M2). For the lactate assay, 20 µl of sample was added to each well. Then, 100 µl of diluted assay buffer, 20 µl of cofactor mixture, and 20 µl of fluorometric substrate were added to all wells being used. The 40 µl of enzyme mixture was added to initiate the reactions. Last, the plate was incubated for 20 min at room temperature and the fluorescence was read using an excitation wavelength between 530–540 nm and emission wavelength between 585–595 nm with microplate reader (Soft Max Pro 4.7.1, Spectra max M2).

## Proteomic analyses

Experiments were conducted in triplicates (n=100 blastocyst for each replicate). Proteins were extracted with 8 M urea, 50 mM ammonium bicarbonate and Benzonase 24 U/100 ml, reduced with TCEP, alkylated with iodoacetamide, and digested overnight with Trypsin/Lys-C mix followed by desalting with C18 cartridges in an automated fashion using an AssayMap BRAVO (Agilent). Peptide mixture was analyzed by LC-MS/MS using a Proxeon EASY nanoLC system (Thermo Fisher Scientific) coupled to an Orbitrap Fusion Lumos mass spectrometer equipped with FAIMS Pro device (Thermo Fisher Scientific). Peptides were separated using an analytical C18 Aurora column (75 μm × 250 mm, 1.6 μm particles; IonOpticks) at a flow rate of 300 nL/min using a 140 min gradient: 1 to 6% B in 1 min, 6 to 23% B in 90 min, 23 to 34% B in 48 min, and 34 to 50% B in 1 min (A=FA 0.1%; B=80% ACN: 0.1% FA). The mass spectrometer was operated in positive data-dependent acquisition mode, and the Thermo FAIMS Pro device was set to standard resolution. A three-experiment method was set up where each experiment utilized a different FAIMS Pro compensation voltage: −50, −70, and –80 Volts, and each of the three experiments had a 1 s cycle time. A high-resolution MS1 scan in the Orbitrap (m/z range 350–1500, 120 k resolution at m/z 200, AGC 4e5 with maximum injection time of 50 ms, RF lens 30%) was collected in top speed mode with 1 s cycles for the survey and the MS/MS scans. For MS2 spectra, ions with charge state between +2 and+7 were isolated with the quadrupole mass filter using a 0.7 m/z isolation window, fragmented with higher-energy collisional dissociation with normalized collision energy of 30%, and the resulting fragments were detected in the ion trap as rapid scan mode with AGC of 5e4 and maximum injection time of 35 ms. The dynamic exclusion was set to 20 s with a 10 ppm mass tolerance around the precursor.

Mass spectra were analyzed with SpectroMine software (Biognosys, version 2.7.210226.47784). Search criteria used were: full tryptic specificity (cleavage after lysine or arginine residues unless followed by proline), missed cleavages were allowed, carbamidomethylation (C) was set as fixed modification and oxidation (M) as a variable modification. The false identification rate was set to 1%. Quality control, relative quantification, and downstream analysis were performed using the artMS Bioconductor package, which uses MSstats for normalization and differential analysis (*Choi et al., 2014*). Pathway enrichment analysis was performed using the PathView Bioconductor package (*Luo and Brouwer, 2013*).

## Western Blotting

Extracts for SDS-PAGE were prepared from whole gonadal fat, muscle (gastrocnemius) or liver tissues of 40 weeks old mice generated in vivo (FB, n=8, 50% females) or by IVF (n=8, 50% females) after culture in 5% oxygen (*Feuer et al., 2014b*). In addition, each individual tissue sample had the blot repeated four times (=4 technical replicates): the average of the 4 measurements was used as the mean intensity for that sample.

Of note, we did not generate a cohort of adult mice by IVF after culture in 20% oxygen. Tissues were homogenized in cold tissue extraction buffer (100 mM Tris, 2 mM Na3VO4, 100 mM NaCl, 1% Triton X-100, 1 mM EDTA, 10% glycerol, 1% EGTA, 0.1% SDS, 1 mM NaF, 0.5% deoxycholate, 20% Na4P2O7, with protease inhibitors). Protein levels were quantified by BCA protein assay (Thermo Scientific). 15 μg tissue were subjected to gel electrophoresis on 12% PAGEr Gold Precast gels (Lonza) and blotted on to Immobilon-FL membranes (Millipore) using semi-dry transfer (Bio-Rad). Dried membranes were reactivated in methanol, rinsed with PBS, and blocked in Odyssey Blocking Buffer (LI-COR Biosciences, 927–40000). Membranes were then probed overnight at 4°C with primary antibodies diluted 1:1000 in blocking buffer with 0.1% Tween-20, washed in PBS and incubated with secondary antibody diluted in blocking buffer containing 0.1% SDS and 0.1% Tween20. Protein signal detection was performed using the LI-COR Odyssey Imaging System.

## Statistical analyses

Experiments were performed in triplicates. Only lactate and pyruvate levels were measured in a single biological replicate with 10 technical replicates, given the large number of embryos needed. All data were analyzed by the One-way analysis of variance with Tukey's multiple comparison test or unpaired t-test using GraphPad Prism 9.0 (GraphPad, San Diego, CA) were applied to analyze the experimental data. All data are expressed as the means values ± SD. Differences were considered statistically significant if $p < 0.05$.

## Acknowledgements

The authors wish to thank Dr. Diane Barber and Dr. Yi for their help in optimizing intracellular pH measurements and Dr George Brooks for the helpful feedback on the manuscript. PFR designed the study. SHL and PFR conceived the experiments. SHL, XL, and DJM conducted the experiments. All authors analyzed the data and results. All authors wrote and reviewed the manuscript. This work was funded by R01 R01HD092267 to PFR. This research was supported by Basic Science Research Program through the National Research Foundation of Korea (NRF) funded by the Ministry of Education (2021R1A6A3A14046145) to SHL.

## Additional information

### Funding

| Funder | Grant reference number | Author |
| --- | --- | --- |
| Eunice Kennedy Shriver National Institute of Child Health and Human Development | R01 R01HD092267 | Paolo F Rinaudo |
| Basic Science Research Program through the National Research Foundation of Korea | 2021R1A6A3A14046145 | Seok Hee Lee |

The funders had no role in study design, data collection and interpretation, or the decision to submit the work for publication.

### Author contributions

Seok Hee Lee, Data curation, Formal analysis, Methodology, Writing - original draft, Writing - review and editing; Xiaowei Liu, Formal analysis, Methodology, Writing - review and editing; David Jimenez-Morales, Methodology, Writing - review and editing; Paolo F Rinaudo, Conceptualization, Resources, Data curation, Formal analysis, Funding acquisition, Investigation, Methodology, Project administration, Writing - review and editing

### Author ORCIDs

Seok Hee Lee ⓘ http://orcid.org/0000-0002-4656-1618
David Jimenez-Morales ⓘ http://orcid.org/0000-0003-4356-6461
Paolo F Rinaudo ⓘ http://orcid.org/0000-0002-6528-6009

### Ethics

Animal experiments were approved by the Institutional Animal Care and Use Committee (#AN181614-03) of the University of California, San Francisco.

### Decision letter and Author response

Decision letter https://doi.org/10.7554/eLife.79153.sa1
Author response https://doi.org/10.7554/eLife.79153.sa2

## Additional files

### Supplementary files
• MDAR checklist

• Source data 1. The source data for *Figures 1–7* and *Figure 1—figure supplement 1*, *Figure 4—figure supplement 1*, *Figure 7—figure supplement 1*.

### Data availability
All data generated or analyzed during this study are included in the manuscript and supporting file.

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
