## [Editor Report]

Some children conceived by assisted reproductive technologies (ART) exhibit metabolic differences compared to those conceived naturally and the causes are unknown. This work reveals possible explanations for the metabolic differences and provides opportunities to improve ART and prevent the differences. This is a valuable contribution and will be of special interest to practitioners of ART, as well as to developmental and reproductive biologists.

---

## [Decision Letter]

**Decision letter after peer review:**

Thank you for submitting your article "Murine blastocysts generated by in vitro fertilization show increased Warburg metabolism and altered lactate production" for consideration by *eLife*. Your article has been reviewed by 3 peer reviewers, one of whom is a member of our Board of Reviewing Editors, and the evaluation has been overseen by Ricardo Azziz as the Senior Editor. The following individual involved in the review of your submission has agreed to reveal their identity: Kelle Moley (Reviewer #2).

Essential revisions:

The authors should consider and comment on the following questions and criticisms from the three referees when revising their manuscript.

1. Many abbreviations are used in this manuscript. The authors should provide a list of abbreviations at the beginning of the manuscript. Also, the authors should provide a definition/description of Warburg metabolism up-front in the Introduction rather than in the Discussion. These are editorial changes.

2. The authors provide measurements and statistics in the Text, Figures, and Legends. To improve the clarity of the Results section (currently awkward to read) it may be best to remove the measurements and statistics since these are in the Figures and Legends and are redundant in the Results. These are editorial changes.

3. All of the measurements compare natural and IVF blastocysts, and do not look at earlier stage embryos to examine when the metabolic changes begin to occur. For example, it would be useful to look at the developmental timeline (2-cell to blastocyst) for the accumulation of ROS.

4. In Figure 2D the authors show the RFU for IVF5% and IVF20% but do not provide an RFU for flushed blastocysts (FB).

5. It is unclear how the authors quantitated the Western blots in Figure 7. The flushed blastocysts (FB) blots, like the IVF blots, exhibit quite a bit of variation in terms of intensity. Since there is so much variability, how did the authors quantitate the differences in LDHB and MCT1 levels in flushed blastocysts (FB) and IVF blots?

6. Is LDH-B the only LDH isozyme in these murine blastocysts? How about human blastocysts? In this context, there should be some discussion of the author's findings with murine blastocysts and previously published comparable findings with human blastocysts (in the Introduction and Discussion).

7. It would be useful to the reader to have some estimate of the oxygen tensions routinely used in human IVF treatments (e.g., how many used 5% and how many used 20% oxygen). Also, can the effects of ROS on IVF blastocyst metabolism be minimized by the addition of other components to the culture medium (e.g., glutathione and/or NAD) Is there metabolic heterogeneity within individual murine blastocysts as has been proposed for humans?

8. Several methods may not be reliable to quantify the parameters analyzed. For example, determining protein content by immunofluorescence may be misleading as it can be affected by multiple parameters.

9. Intracellular pH was also analyzed by an assay also based on immunofluorescence. This measurement could be affected by embryo size (the blastocoel is a call-devoid cavity).

10. Given the small size of these embryos (~80 µm diameter), it is unclear how they alter significantly the composition of 500 µl of culture medium.

11. Performance of this work in a mouse model could be considered a weakness. However, due to the lack of appropriate cell lines or access to large numbers of human embryos, mouse embryos recapitulate the timing of the early developmental period in humans and are experimentally preferred due to the large number of embryos obtained by superovulation.

[Editors’ note: further revisions were suggested prior to acceptance, as described below.]

Thank you for resubmitting your work entitled "Murine blastocysts generated by in vitro fertilization show increased Warburg metabolism and altered lactate production" for further consideration by *eLife*. Your revised article has been evaluated by Ricardo Azziz (Senior Editor) and two original reviewers.

We are nearly ready to proceed with acceptance, but first could you please respond (in a rebuttal and/or in the text) to the three points raised by Reviewer #3?

*Reviewer #2 (Recommendations for the authors):*

The authors have adequately answered my questions. The changes they have made in response to the other reviewers comments as well as mine have greatly improved the manuscript.

*Reviewer #3 (Recommendations for the authors):*

Authors have clarified some relevant points undisclosed or wrongly described in the first version, such as embryo collection timing and that group culture in a small amount of medium was used to collect samples for metabolite analysis. However, no further experiments were conducted, so I still believe that conclusions are not well supported by the data. Anyway, if other reviewers do not consider these points as critical, we can proceed with manuscript acceptance.

(1) I understand that efforts to obtain in vivo and in vitro blastocyst at a similar stage were taken, but there is still a pretty significant difference in embryo size (50 vs. 70 cells) which can bias the analysis. In other words, some the differences observed may be due to developmental stage rather than in vivo vs. in vitro, as embryo metabolism changes drastically in the morula to blastocyst transition.

(2) Alternative methods to fluorescence-based microscopy analysis are available to confirm results.

(3) It would be advisable to support the hypothesis of lines 333-344 by epigenetics analysis of in vitro vs. in vivo data conducted by the authors or publicly available.

---

## [Author Response]

Essential revisions:The authors should consider and comment on the following questions and criticisms from the three referees when revising their manuscript.1. Many abbreviations are used in this manuscript. The authors should provide a list of abbreviations at the beginning of the manuscript. Also, the authors should provide a definition/description of Warburg metabolism up-front in the Introduction rather than in the Discussion. These are editorial changes.

A list of abbreviation has been provided at the beginning of the manuscript, as follow

ART Assisted reproductive technologies

ATP Adenosine triphosphate

CMF2HC 4-chloromethyl- 6,8- difluoro-7-hydroxycoumarin

COCs Cumulus-oocyte complexes

DCF Dichlorodihydrofluorescein

DNPH 2,4-Dinitrophenylhydrazine

ECAR Extracellular acidification rate

FB Flushed blastocyst

FCCP Carbonyl cyanide-4-trifluoromethoxyphenylhydrazone

GSH Glutathione

hCG Human chorionic gonadotropin

HTF Human tubal fluid

H2DCFDA 2′,7′-dichlorodihydro-ﬂuorescein diacetate

IVC in vitro cultivation

IVF in vitro fertilization

KSOM Potassium simplex optimization medium

LDHA Lactate dehydrogenase A

LDHB Lactate dehydrogenase B

MCT1 Monocarboxylate transporter 1

MPA Metaphosphoric acid

NAD Nicotinamide adenine dinucleotide

OCR Oxygen consumption rate

PBS Phosphate buffer saline

PER Proton efflux rate

PGF2 Prostaglandin F2

PMSG Pregnant mare serum gonadotrophin

PPP Pentose phosphate pathway

PVA Poly vinyl alcohol

ROS Reactive oxygen species

TCA Tricarboxylic cycle

2-DG 2-deoxy-D-glucose

8-OHdG 8-Hydroxyguanosine

Further, the definition of Warburg metabolism is currently provided in the Introduction (and removed from the discussion) as follows:

Line 89-96: “Preimplantation embryo metabolism is unique and different from somatic cells. The Krebs cycle is the main source of energy throughout the preimplantation period. Glycolysis is slowly utilized during the first 1-2 days of development but both an increase in glycolysis and oxygen consumption (via mitochondria) is notable at the blastocyst stage (Leese, 2012). Importantly, preimplantation embryos show an accentuation of the Warburg metabolism (Warburg, 1956), i.e. higher production of ATP through glycolysis and increased conversion of pyruvate into lactate, even in the presence of oxygen (Redel 2011).”

2. The authors provide measurements and statistics in the Text, Figures, and Legends. To improve the clarity of the Results section (currently awkward to read) it may be best to remove the measurements and statistics since these are in the Figures and Legends and are redundant in the Results. These are editorial changes.

Done: we have removed the measurements and statistics in the Results section to improve clarity.

3. All of the measurements compare natural and IVF blastocysts, and do not look at earlier stage embryos to examine when the metabolic changes begin to occur. For example, it would be useful to look at the developmental timeline (2-cell to blastocyst) for the accumulation of ROS.

Thank you for this valuable insight. We decided to focus on the blastocyst stage, since this is the most common stage at which human embryos are transferred and therefore the blastocyst metabolic “state” is clinically relevant. We believe that a cumulative increase in metabolic and oxidative stress is very likely with longer embryo culture and therefore mapping the metabolic state through preimplantation embryos would provide valuable information. We have added this insight in the discussion

Line 326-329: “In this work we have focused on studying the blastocyst stage, since this is the stage at which human embryos are most commonly transferred. Future studies should examine how metabolic differences, ROS and oxidative damage occur during earlier stages of embryo development, both in mice and possibly human embryos.”

4. In Figure 2D the authors show the RFU for IVF5% and IVF20% but do not provide an RFU for flushed blastocysts (FB).

We believe the reviewer is referring to “Figure 1D:Extracellular ROS level” (that does not have value for FB group) as opposed to “Figure 2D” levels of PGF2α, (that has value for FB).

Figure 1D shows extracellular ROS level of the media where embryos were cultured for 4 days (in either 5% or 20% oxygen).The extracellular ROS level were measured in the culture medium, which can be only analyzed in the in vitro sample. ROS

This has been clarified in the legend of Figure 1:

Line 622-623: “Since FB were not cultured, the ROS value for the FB group is missing”

5. It is unclear how the authors quantitated the Western blots in Figure 7. The flushed blastocysts (FB) blots, like the IVF blots, exhibit quite a bit of variation in terms of intensity. Since there is so much variability, how did the authors quantitate the differences in LDHB and MCT1 levels in flushed blastocysts (FB) and IVF blots?

We also noted a variation in intensity between the samples. To control for this, we performed a large number of replicates.

For each tissue, unless differentially described, we compared 8 IVF 5% samples vs. 8 FB samples (=8 biological replicates). In addition, each individual tissue samples had the blot repeated 4 times ( = 4 technical replicates): the average of the 4 measurements was used as the mean intensity for that sample.

This has been added to the method section,

Line 576-578: “In addition, each individual tissue sample had the blot repeated 4 times ( = 4 technical replicates): the average of the 4 measurements was used as the mean intensity for that sample.”

6. Is LDH-B the only LDH isozyme in these murine blastocysts? How about human blastocysts? In this context, there should be some discussion of the author's findings with murine blastocysts and previously published comparable findings with human blastocysts (in the Introduction and Discussion).

The enzyme Lactate dehydrogenase catalyzes the inter-conversion of pyruvate and L-lactate with concomitant inter-conversion of NADH and NAD+. There are 2 main mammalian proteins LDH-A and LDH B which in humans are encoded by the LDH-A and LDH-B genes. These enzymes are ubiquitously expressed in tissue. Two additional mammalian LDH subunits can be found in specific compartments: LDHC is a testes-specific LDH protein, (encoded by the LDH-C gene) and LDH-Bx, is peroxisome-specific.

Both LDH-A and B isoenzyme are active in the preimplantation mouse embryos, but LDH-B is the principal form. LDH-A will become predominant at the time of implantation (Auerbach 1967; Lane 2000). Human studies suggest a similar pattern: the mRNA of LDH-B was detected in human oocytes, 4- and 8-cell embryos, while the mRNA for LDH-A was detected only in two of four oocytes and one of three 8-cell embryos. However, the mRNA for testis-specific LDH-C was not detected (Li 2006).

This information has been added to the manuscript as follows:

Introduction line 95-96: “This process is facilitated by the lactate dehydrogenase class of enzymes (LDH-A and LDH-B) (Redel 2011).”

Line 309-314: “Both LDH-A and B isoenzyme are active in the preimplantation mouse embryos, but LDH-B is the principal form. LDH-A will become predominant at the time of implantation (Auerbach 1967; Lane 2000). Human studies suggest a similar pattern: the mRNA of LDH-B was detected in human oocytes, 4- and 8-cell embryos, while the mRNA for LDH-A was detected only in two of four oocytes and one of three 8-cell embryos. However, the mRNA for testis-specific LDH-C was not detected (Li 2006).”

7. It would be useful to the reader to have some estimate of the oxygen tensions routinely used in human IVF treatments (e.g., how many used 5% and how many used 20% oxygen). Also, can the effects of ROS on IVF blastocyst metabolism be minimized by the addition of other components to the culture medium (e.g., glutathione and/or NAD) Is there metabolic heterogeneity within individual murine blastocysts as has been proposed for humans?

Although is preeminently clear that low oxygen tension (~5% oxygen) should be used to culture mammalian embryos, a survey performed in 2014 revealed that out of 265 IVF clinics from 54 different countries, less than 25% used physiological (~5%) oxygen to culture human embryos (Christianson 2014).

Addition of antioxidants (resveratrol, Epigallocatechin-3-gallate, Lysophosphatidic acid, etc) to embryo culture media, has been studied in animal models by numerous investigators, but currently there is no consensus on the optimal combination of antioxidant to use. Interestingly, ROS production was documented after culture of embryos using commercially available media, even under conditions of low oxygen concentration (Martin-Romero 2008).

The question of metabolic heterogeneity in individual murine blastocysts is very important. Our results indicate that individual blastocysts show variation in intracellular pH, oxidative damage, ROS, NAD levels. Given this variability, and based on the Goldilocks principal of embryo health (Leese 2016), embryo showing less stress could be identified and selected for early transfer.

This information has been added to the manuscript as follows:

Line 76-78: “In particular, either 5% (physiologic) or 20% oxygen (atmospheric) concentrations have been widely used in clinical settings and in 2014 only 25% of 265 IVF clinics reported of using 5% oxygen for culture of human embryos (Christianson 2014).”

Line 278-281: “Given that individual blastocysts show variation in intracellular pH, oxidative damage, ROS, NAD levels, and based on the Goldilocks principal of embryo health (Leese 2016), embryo showing less stress could be identified and selected for early transfer.”

Line 329-332: “Although ROS production was documented after culture of embryos using commercially available media, even under conditions of low oxygen concentration (Martin-Romero 2008), design of optimized culture media, aimed at reduce ROS generation should be encouraged.”

8. Several methods may not be reliable to quantify the parameters analyzed. For example, determining protein content by immunofluorescence may be misleading as it can be affected by multiple parameters.

We appreciate the comments and concerns. Any single method can result in error and possible bias. Immunofluorescence analysis is a robust method that has been used to analyze the distribution of proteins in cells or tissues. For instance, oxidative stress (Liu et al., 2022, Reprod Domest Anim), several signaling molecule (Spirkova et al., 2022, Biol Reprod) and DNA methylation level (Diaz et al., 2021, Fron Gent) have been measured by immunofluorescence in preimplantation embryos and oocytes. It our study, to minimize errors, we followed exactly the same protocol and we found immunofluorescence to be reliable. In addition, global proteomics analysis of blastocysts provide partial independent confirmation of our results. While LDH-A and MCT1 were not detected, LDH-B was detected and found to be lower in IVF blastocysts, exactly as show by IF studies. Finally, western blot analysis of adult tissues confirmed reduction in LDH-B and MCT-1 levels.

These comments have been added to the discussion as follows:

Line 299-302: “Unsupervised global proteomics analysis revealed that LDH-B was downregulated in IVF embryos. We confirmed these results by performing immunofluorescence studies. In addition we found that IVF embryos showed downregulation of both LDHA and B and of the monocarboxylate transporter, MCT 1, providing an explanation for the increase in their lactate levels”

9. Intracellular pH was also analyzed by an assay also based on immunofluorescence. This measurement could be affected by embryo size (the blastocoel is a call-devoid cavity).

Thank you for the comment. To control for the possible variation in blastocoel size in different embryos, we compared immunofluorescence level of only the inner cell mass and trophoblast region of blastocysts and excluded the blastocoel region.

This clarification has been added to the method section as follow:

Line 488-491: “To control for the possible variation in blastocoel size in different embryos, we compared immunofluorescence level of only the inner cell mass and trophoblast region of blastocysts and excluded the blastocoel region.”

10. Given the small size of these embryos (~80 µm diameter), it is unclear how they alter significantly the composition of 500 µl of culture medium.

To collect 300 blastocysts, we performed multiple IVF, each IVF resulting in 10-20 blastocysts cultured in 30 microliters of media. While intracellular lactate and pyruvate were performed on the embryos collected, the media from different experiments was pooled to a final 500 microliter volume. Lactate and pyruvate levels were measured in this final volume for each group of embryo (FB, IVF5% and IVF20%)

This has been clarified in the method section as follows:

Line 516-519: “To collect 300 blastocysts, we performed multiple IVF, each IVF resulting in 10-20 blastocysts cultured in 30 microliters of media. While intracellular lactate and pyruvate were performed on the embryos collected, the media from different experiments was pooled to a final 500 microliter volume.”

11. Performance of this work in a mouse model could be considered a weakness. However, due to the lack of appropriate cell lines or access to large numbers of human embryos, mouse embryos recapitulate the timing of the early developmental period in humans and are experimentally preferred due to the large number of embryos obtained by superovulation.

Thank you for the comment. We agree that mouse embryo can be used for recapitulating the timing of the early development in humans, but confirmation in humans would be important.

We have added this in the discussion:

Line 327-329: “Future studies should examine how metabolic differences, ROS and oxidative damage occur during earlier stages of embryo development, both in mice and possibly human embryos.”

[Editors’ note: further revisions were suggested prior to acceptance, as described below.]

We are nearly ready to proceed with acceptance, but first could you please respond (in a rebuttal and/or in the text) to the three points raised by Reviewer #3?Reviewer #3 (Recommendations for the authors):Authors have clarified some relevant points undisclosed or wrongly described in the first version, such as embryo collection timing and that group culture in a small amount of medium was used to collect samples for metabolite analysis. However, no further experiments were conducted, so I still believe that conclusions are not well supported by the data. Anyway, if other reviewers do not consider these points as critical, we can proceed with manuscript acceptance.(1) I understand that efforts to obtain in vivo and in vitro blastocyst at a similar stage were taken, but there is still a pretty significant difference in embryo size (50 vs. 70 cells) which can bias the analysis. In other words, some the differences observed may be due to developmental stage rather than in vivo vs. in vitro, as embryo metabolism changes drastically in the morula to blastocyst transition.

Thank you for the comment. We agree with the reviewers that the differences in cell numbers are noteworthy. As shown in Figure 1, Figure Supplement 1 we and other invesitgators in the field of reprogramming (Doherty 2000; Ecker 2004; Vrooman 2022; Watkins 2007), have selected and analyzed blastocysts with similar morphology and developmental stage (expanded blastocysts). This is the selection criteria utilized by human embryologist to select an embryo to transfer. Rather than indicating different stages of development, we interpret the differences in total cell number as indicating the profound effects of different embryo environments (in vivo vs. in vitro culture, 5% vs. 20% Oxygen concentration, or type of media; for example Whittens vs KSOM, etc…) on preimplantation embryo development. In other words, the different cell number is a marker of the stress to which embryos have been exposed. If we had waited longer for in vitro embryos to grow in culture, in vitro embryo would have started to hatch and therefore would not have had similar morphology to the in vivo embryos.

In summary, we observe that different culture conditions alter embryo development (cell number, gene expression, epigenetic marks…) to such an extent that embryos cultured in vitro might be predisposed to adult onset diseases (Fleming 2018).

(2) Alternative methods to fluorescence-based microscopy analysis are available to confirm results.

Thank you for your comment. We agree that the alternative method including western blot could be available to confirm our fluorescence-based microscopy results. However, in line with many other authors (Saiz 2016; Schuh 2007), we believe the immunofluorescence analysis in preimplantation embryos and oocytes is of sufficient robustness to draw reliable conclusions. In addition, our global proteomics analysis of blastocysts and western blot analysis of adult tissues offer a measure of support to the immunofluorescence results. In fact, unsupervised proteomics revealed that LDH-B was decreased in IVF-generated embryos. We have modified the manuscript as follow:

Line 321-323: In addition, a possible limitation of the paper has been the prevalent utilization of fluorescence-based microscopy as a method to identify differences in expression of oxidative marks or protein levels.

(3) It would be advisable to support the hypothesis of lines 333-344 by epigenetics analysis of in vitro vs. in vivo data conducted by the authors or publicly available.

Thank you for the suggestions. We have added the work of Yang et al., (Yang 2021), as a partial support to our hypothesis. In this work the authors found that hypoxic embryo culture (2% Oxygen) results in lower level of histone lactylation (H3K23la, H3K18la). We have modified the discussion as follow:

Line 329-330: “A recent study (Yang 2021) showed that hypoxic embryo culture alters histone lactylation marks (H3K23la, H3K18la), providing partial support to this hypothesis.”

References

Doherty, AS, MR Mann, KD Tremblay, MS Bartolomei, and RM Schultz 2000 Differential effects of culture on imprinted H19 expression in the preimplantation mouse embryo. *Biol Reprod* 62 1526-1535.

Ecker, DJ, P Stein, Z Xu, CJ Williams, GS Kopf, WB Bilker, T Abel, and RM Schultz 2004 Long-term effects of culture of preimplantation mouse embryos on behavior. Proceedings of the National Academy of Sciences of the United States of America 101 1595-1600.

Fleming, TP, AJ Watkins, MA Velazquez, JC Mathers, AM Prentice, J Stephenson, M Barker, R Saffery, CS Yajnik, JJ Eckert, MA Hanson, T Forrester, PD Gluckman, and KM Godfrey 2018 Origins of lifetime health around the time of conception: causes and consequences. *Lancet* 391 1842-1852.

Saiz, N, M Kang, N Schrode, X Lou, and AK Hadjantonakis 2016 Quantitative Analysis of Protein Expression to Study Lineage Specification in Mouse Preimplantation Embryos. *J Vis Exp* 53654.

Schuh, M, and J Ellenberg 2007 Self-organization of MTOCs replaces centrosome function during acentrosomal spindle assembly in live mouse oocytes. *Cell* 130 484-498.

Vrooman, LA, EA Rhon-Calderon, KV Suri, AK Dahiya, Y Lan, RM Schultz, and MS Bartolomei 2022 Placental Abnormalities are Associated With Specific Windows of Embryo Culture in a Mouse Model. *Front Cell Dev Biol* 10 884088.

Watkins, AJ, D Platt, T Papenbrock, A Wilkins, JJ Eckert, WY Kwong, C Osmond, M Hanson, and TP Fleming 2007 Mouse embryo culture induces changes in postnatal phenotype including raised systolic blood pressure. *Proc Natl Acad Sci U S A* 104 54495454.

Yang, W, P Wang, P Cao, S Wang, Y Yang, H Su, and B Nashun 2021 Hypoxic in vitro culture reduces histone lactylation and impairs pre-implantation embryonic development in mice. *Epigenetics Chromatin* 14 57.